# Understanding the Staged Dynamics of Transformers in Learning Latent Structure

**Anonymous Authors[1]**

## Abstract

Language modeling has shown us that transformers can discover latent structure from context, but the dynamics of how they acquire different components of that structure remain poorly understood, leading to assertions that models just remix training data. In this work, we use the Alchemy benchmark in a controlled setting (Wang et al., 2021) to investigate latent structure learning. We train a small decoder-only transformer on three task variants: 1) inferring missing transitions from partial contextual information, 2) composing simple rules to solve multi-transition sequences, and 3) decomposing complex multi-step examples to infer intermediate transitions. By factorizing each task into interpretable components, we show that the model learns the different latent structure components in discrete *stages*. We also observe an asymmetry: the model composes fundamental transitions robustly, but struggles to decompose complex examples to discover the atomic transitions. Finally, using causal interventions, we identify layer-specific plasticity windows during which freezing substantially delays or prevents stage completion. These findings provide insight into *how* a transformer model acquires latent structure, offering a detailed view of how capabilities evolve during training.

## 1. Introduction

In the context of language modeling, it is clear that transformers can learn latent structure. For example, models learn syntactic hierarchies without supervision, with attention heads encoding structures such as dependency trees (Hewitt & Manning, 2019; Manning et al., 2020; Ravishankar et al., 2021), and encode causal graphs in their attention mechanisms (Nichani et al., 2024). However, because of the complexity of latent structures in language, it is difficult to study the progression of learning latent structure and even more difficult to show that models are truly generalizing and not just remixing training data (Bender et al., 2021).

Here, we explore a setting with a clear and controlled latent structure: the Alchemy benchmark (Wang et al., 2021). The latent structures in Alchemy allow us to study the learning progression in multiple settings by carefully controlling what the model observes during training. We also control the data split so that learning the latent structure is the only way to reach perfect accuracy. In many settings, we observe staged learning dynamics: performance improves in plateaus, followed by sudden jumps, suggesting that models acquire capabilities in stages rather than continuously. Although previous studies have reported staged learning in many settings (Wei et al., 2022a; Singh et al., 2024; Chen et al., 2024; Song et al., 2025; Gopalani & Hu, 2025), the complexity of these settings makes it difficult to attribute the stages to what is actually learned. Our analysis allows us to attribute the stages to specific aspects of Alchemy's inherent latent structure.

In the Alchemy setting, players start with a stone whose properties can be altered by applying various potions, changing the stone's features accordingly. The set of stones and the transitions between them, induced by potions, form a "chemistry". Each chemistry is represented as a directed graph with eight vertices arranged in a cubic structure (Figure 1, left panel). We investigate latent structure learning dynamics by withholding observations of latent features, and increasing task complexity. These controlled experiments demonstrate that rapid performance improvements align with the successful learning of the different aspects of the chemistry.

We train a small decoder-only transformer model ($\sim$ 2M parameters) (Vaswani et al., 2017) on three formulations of the Alchemy task (Figure 1): 1) We strategically withhold information about the effects of specific potions and test the model's ability to infer the effects of these missing potions (Figure 1, right callout), 2) We investigate compositionality by providing the model with all fundamental, single potion effects and test its ability to compose these single potions

[1]Anonymous Institution, Anonymous City, Anonymous Region, Anonymous Country. Correspondence to: Anonymous Author <anon.email@domain.com>.

Preliminary work. Under review by the International Conference on Machine Learning (ICML). Do not distribute.

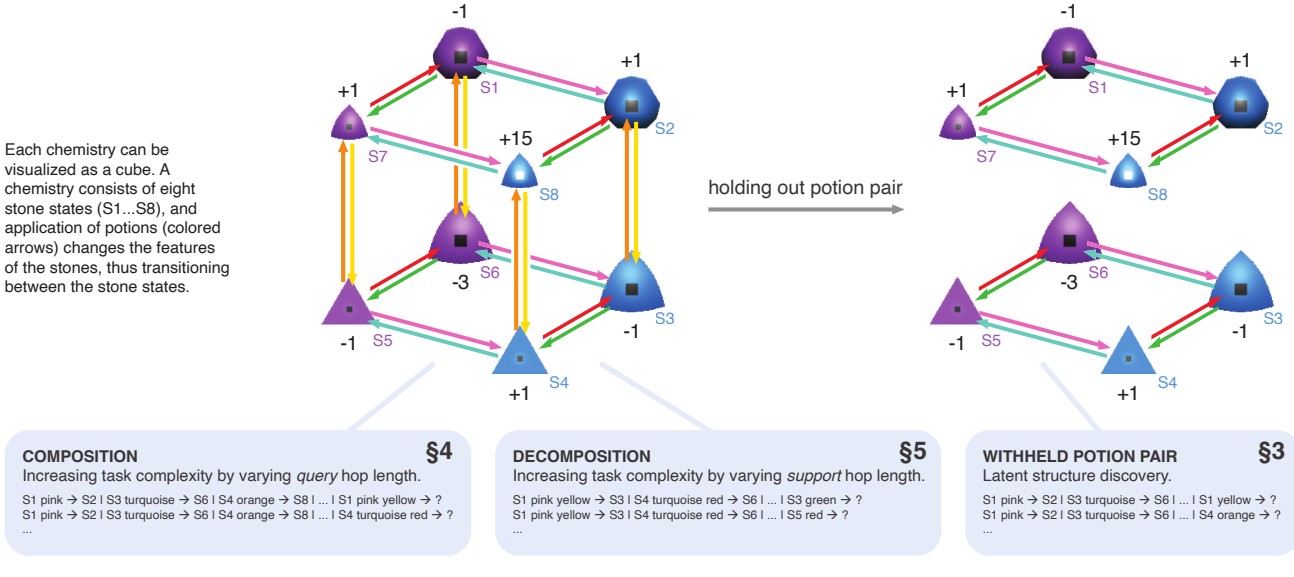

Each chemistry can be visualized as a cube. A chemistry consists of eight stone states (S1...S8), and application of potions (colored arrows) changes the features of the stones, thus transitioning between the stone states.

holding out potion pair

**COMPOSITION** §4
Increasing task complexity by varying *query* hop length.
S1 pink → S2 | S3 turquoise → S6 | S4 orange → S8 | ... | S1 pink yellow → ?
S1 pink → S2 | S3 turquoise → S6 | S4 orange → S8 | ... | S4 turquoise red → ?
...

**DECOMPOSITION** §5
Increasing task complexity by varying *support* hop length.
S1 pink yellow → S3 | S4 turquoise red → S6 | ... | S3 green → ?
S1 pink yellow → S3 | S4 turquoise red → S6 | ... | S5 red → ?
...

**WITHHELD POTION PAIR** §3
Latent structure discovery.
S1 pink → S2 | S3 turquoise → S6 | ... | S1 yellow → ?
S1 pink → S2 | S3 turquoise → S6 | ... | S4 orange → ?
...

*Figure 1.* Overview of Alchemy chemistry structure and experimental tasks. A chemistry graph (left panel) contains eight stone states (vertices) connected by potion-induced transitions (edges), that change stone features. Right callout: experiment to investigate latent structure learning dynamics. For a chemistry, transitions for a randomly selected potion pair are withheld (e.g., yellow/orange); the model needs to correctly infer their effects. To solve this task, the model needs to correctly align the two disconnected halves created from withholding the potion pair (right panel). Left callout: composition task where the model needs to solve a multi-hop query given all 1-hop (single-step) support transitions. Middle callout: decomposition requires solving a 1-hop query given all multi-hop transitions in the support , given a hop length. In our experiments, we fully enumerate the stone features in the support and query. The part after the last | is the query. Numbers in the callouts denote the respective sections of the paper discussing results. Figure adapted from Wang et al. (2021).

to solve novel, multi-step potion sequences (Figure 1, left callout), 3) We test the model's decomposition ability by providing it with multi-step potion sequences and requiring it to infer the hidden single-step potion effects (Figure 1, middle callout). Finally, we use freezing interventions to causally infer layer-specific plasticity requirements for latent structure acquisition.

Our central contribution is to analyze staged dynamics to deeply understand latent structure learning in a controlled setting. We factorize the model's overall accuracy into interpretable components, corresponding to distinct latent properties, showing that the observed plateaus and jumps in the model performance are not random, but correspond to acquiring specific interpretable sub-skills. Concretely:

1. In the withheld potion effect discovery task, the model infers the effect of a missing potion pair in distinct stages: it first narrows the set of possible outcomes, then gradually learns the correct target (Section 3).

2. When composing single-step potion transitions to solve multi-step potion sequences, the model exhibits invariance to task complexity. Stages are learned at approximately the same time during training, irrespective of the number of steps in a potion sequence (Section 4).

3. When decomposing multi-step potion sequences to infer a single-step potion transition, model convergence

is delayed as sequence length increases, reflected in the delayed learning of later stages (Section 5).

4. Using causal interventions, we identify layer-specific *plasticity windows*: specific periods during which a layer must remain trainable to support staged learning. The plasticity requirement, while distributed across layers, is temporally bounded (Section 6).

We release our code to encourage further research on staged dynamics.

## 2. Methods

In this section, we introduce the Alchemy dataset, its latent structures, and properties. We then introduce notation to motivate the experiments. Finally, we describe our transformer model architecture and the training details.

### 2.1. Dataset

We use the Alchemy dataset (Wang et al., 2021), a compositional meta-reinforcement learning benchmark designed to test a model's ability to discover and exploit latent structures. In Alchemy, players apply potions to a stone, changing its state. Interactions between the stones and potions are defined by a chemistry, and chemistries differ between rounds. All chemistries share a latent cubic graph structure that describes the interactions (Figure 1, left panel).

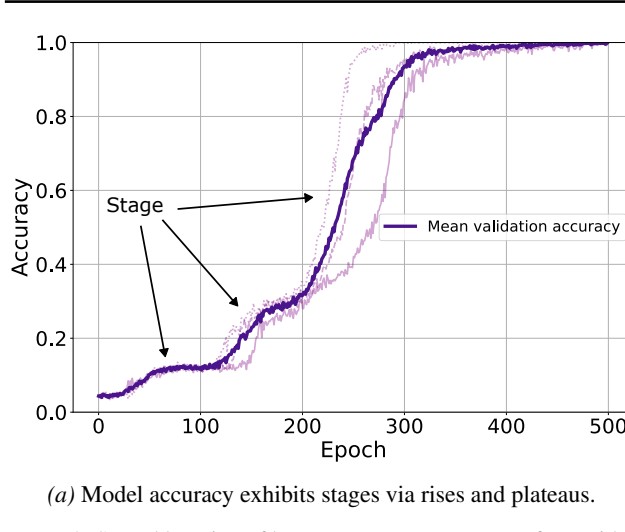

*(a)* Model accuracy exhibits stages via rises and plateaus.

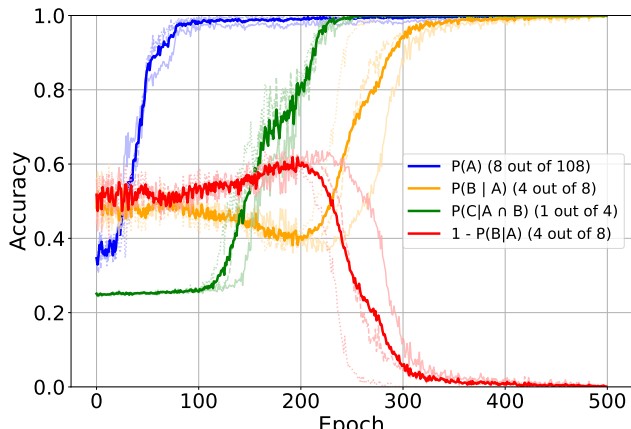

*(b)* Task factorization reveals staged learning dynamics.

*Figure 2.* Staged learning of latent structure components for a withheld potion pair (500 epochs). (a) Overall model performance, where the model learns through discrete stages indicated by rises and plateaus in accuracy. (b) Factorization into components: $\mathbb{P}[A]$ (blue, in-support), $\mathbb{P}[B|A]$ (orange, correct half), and $\mathbb{P}[C|A \cap B]$ (green, exact match). Red shows incorrect half accuracy. $\mathbb{P}[A]\,\mathbb{P}[B \mid A]\,\mathbb{P}[C \mid A \cap B]$ recovers the overall accuracy in Figure 2a. Lighter and darker curves denote individual runs and the mean respectively.

To solve Alchemy, the model cannot memorize the chemistry specific transitions between the stones from a cubic graph structure; instead, it must learn to discover and exploit the invariant latent structures that are consistent across all chemistries. While the original alchemy dataset contains chemistries with missing edges (incomplete graphs), we only consider complete chemistries[1], where each vertex has three outgoing edges.

The chemistries have the following invariances:

- Each chemistry graph has 8 possible stone states positioned on the vertices of the cubic structure. Each stone is defined by four features: color, size, roundness, and reward. The first three perceptual features ('color', 'size', 'roundness') have three possible values. The 'color' feature can take values pink, violet, blue, 'size' can take values small, medium, large, and 'roundness' can take values pointy, medium_round, and round. The 'reward' feature can take one of four values (-3, -1, +1, +15). This creates 108 unique possible stones across all chemistries, and a chemistry contains exactly 8 stones.

- Potions come in fixed complementary pairs (red/-green, yellow/orange, pink/turquoise) that are consistent across all chemistries. Applying a potion changes the stone's features, moving the state as dictated by an edge connecting two vertices, and complementary potions always have inverse effects. For any given chemistry, each potion color corresponds to a single latent transition rule that can be applied to a vertex.

Crucially, a potion's transition rule corresponding to one axis may not align with a single perceptual feature, and thus a single potion can cause multiple perceptual features to change at once. For example, in the chemistry shown in Figure 1 (left panel), the yellow potions adjust the roundness and size by a single level, but keep the color unchanged. By contrast, the pink potion leaves the stone's size and roundness unchanged, but shifts its color by two levels (pink $\rightarrow$ violet $\rightarrow$ blue).

- The reward feature follows a structured distribution consistent across all chemistries: +15 is adjacent only to +1, -3 is adjacent only to -1, +1 is adjacent to +15 and -1, and -1 is adjacent to +1 and -3. Thus, applying a potion changes the reward feature based on the stone's position in the chemistry, with complementary potions inducing inverse reward transitions. Unlike in reinforcement learning settings, where the objective is to maximize reward, here the reward is simply a stone feature.

### 2.2. Notation and Task Formulation:

We frame the Alchemy task as a supervised learning problem. We provide the model with a set of **support** (contextual) examples from a specific chemistry, using which the model must respond to a **query** example. We formalize the notions of **support** and **query** below.

Given a complete chemistry graph, the **support** $S$ consists of triplets with $M$ example transitions: $S = \{(x_{s_m}, z_{s_m}, y_{s_m})\}_{m=1}^{M}$. Each support example represents a transformation in the chemistry, where $x_{s_m}$ is the *start stone state*, $z_{s_m}$ is the *sequence of potions applied*, and $y_{s_m}$ is the resulting *end stone state*. We randomize the order of the

---

[1]Since incomplete chemistries are subgraphs of complete ones, we exclude them to ensure strict disjointness between train and validation sets.


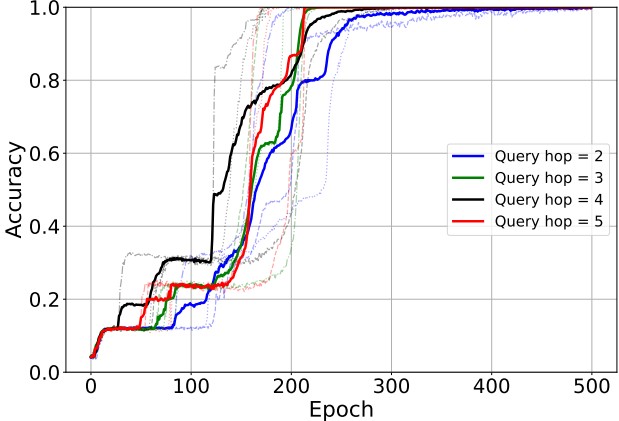

*Figure 3.* Overall composition performance. No noticeable difference in convergence exists with varying $hl_{query} \in \{2, 3, 4, 5\}$, with all hops reaching convergence within 500 epochs. Lighter and darker curves denote individual runs and the mean respectively.

examples in $S$ to prevent any ordering biases. $Q = \{q_i\}_{i=1}^N$ is the set of all possible queries for a chemistry, where each $q_i = (x_i, z_i)$. We ensure that $S$ and $Q$ are disjoint. We train the model to learn $p(y_i \mid q_i, S)$, i.e., given a query $q_i$, the model must predict the correct output stone $y_i$, using the support $S$ to infer the latent chemistry.

To systematically vary task complexity, we introduce 'hop length' ($hl$): the length of the potion sequence $z$. For instance, a 2-hop example ($hl = 2$) corresponds to applying two potions in succession to move from the start stone state to the end stone state in the chemistry (Figure 1). We vary $hl$ for both support ($hl_{support}$), and query ($hl_{query}$).

We use this task formulation to design three systematic experiments (callouts in Figure 1). 1) **Withheld potion pair**: we vary the content of $S$ with $hl_{support} = 1$ but withhold all examples for a randomly selected potion pair. The query $q_i$ with $hl_{query} = 1$ then tests the model's ability to infer the effect of the withheld potions (right callout). (2) **Composition**: we test the model's ability to compose 1-hop support transitions ($hl_{support} = 1$), to solve multi-hop queries ($hl_{query} \in \{2, 3, 4, 5\}$) (left callout). (3) **Decomposition**: we test the model's ability to decompose complex multi-hop support examples ($hl_{support} \in \{2, 3, 4, 5\}$) to solve a 1-hop query ($hl_{query} = 1$) (middle callout).

## 2.3. Training Details

We use a transformer model (Vaswani et al., 2017) with $\sim$ 2M parameters. We use 4 hidden layers, embedding dimension of 256, feedforward dimension of 512, a dropout rate of 0.1, and causal attention. Based on prior work studying learning dynamics (Gopalani & Hu, 2025), we restrict our analyses to a small-scale model.

Following prior work on learning from contextual examples

(Garg et al., 2022; Chan et al., 2022; Singh et al., 2023; Lake & Baroni, 2023), we concatenate all the support examples $S$ in the model input. We use standard cross-entropy loss on the final token prediction. We use a 90 / 10 train - validation split and ensure that validation chemistries are unseen during training. We train all models for 1000 epochs, with a batch size of 32, and with three random initializations. The hyperparameter tuning details appear in Appendix A. As our focus is on learning dynamics rather than optimization failures, we analyze only those runs reaching near-perfect accuracy within the 1000 epoch compute budget.

## 3. Learning the Withheld Potion Pair

In this experiment, we study the learning dynamics of latent structure discovery, wherein the model must infer the effect of a withheld potion pair using the support transitions of the remaining potion pairs. Intuitively, withholding transitions for a potion pair creates two *disconnected* halves of the chemistry graph (Figure 1, right panel). To correctly predict the query output, the model must correctly align the halves relative to one another. This contrasts with previous work where models may rely on surface-level patterns or counting statistics (Mittal et al., 2025). Instead, our setup requires learning the underlying transition rules, preventing models from exploiting superficial statistical shortcuts.

### 3.1. Analytical Study of Learning Stages

We show the results in Figure 2a, which shows multiple stages in the accuracy curve. We systematically study these stages by factorizing the task into interpretable events.

**Sets and Events:** Consider the example chemistry ($chem$) in Figure 1 (left panel), from which we withhold one potion pair (Figure 1, right panel). For a given chemistry $chem$, let $T_{chem}$ be the set of eight stones in the support $S$, containing the data for the remaining transitions. Withholding a potion pair creates two disconnected halves, one of them containing the query stone $x_i$, and the other containing the ground truth $y_i$. We denote the half containing $x_i$ as $H_{chem}^{(same)}$ and the half containing $y_i$ as $H_{chem}^{(other)}$. Solving the task requires narrowing the predictions along these constraints. Given a chemistry $chem$, query $q_i$, and prediction $\hat{y}_i$, we define:

$$A := \{\hat{y}_i \in T_{chem}\} \quad \text{(in-support)},$$
$$B := \{\hat{y}_i \in H_{chem}^{(other)}\} \quad \text{(correct half)},$$
$$C := \{\hat{y}_i = y_i\} \quad \text{(exact match)}.$$

These events leverage meaningful structural properties of the task and represent *sub-skills* that the model must achieve to solve the task. For instance, $A$ is a meaningful event because the correct answer ($y_i$) is always in $T_{chem}$, reflecting whether the model has learned to restrict its predictions from the global space of 108 stones to the current chemistry.

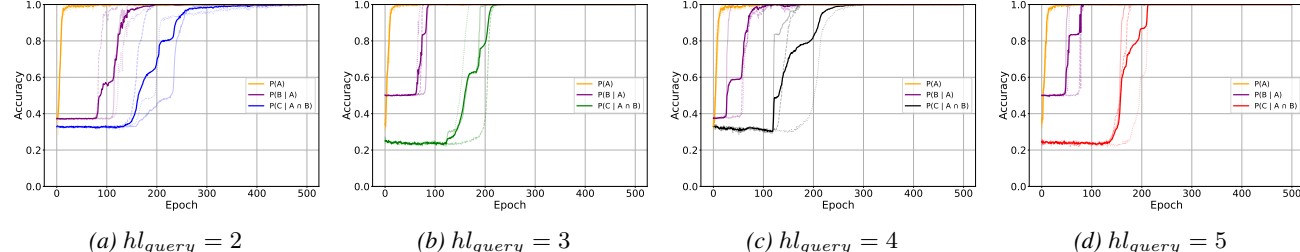

*(a) $hl_{query} = 2$*   *(b) $hl_{query} = 3$*   *(c) $hl_{query} = 4$*   *(d) $hl_{query} = 5$*

*Figure 4.* Staged learning dynamics for composition. We plot $\mathbb{P}[A]$ (orange, in-support), $\mathbb{P}[R \mid A]$ (purple, within reachable stones), and $\mathbb{P}[C \mid A \cap R]$ (exact match) shown as blue, green, black, and red, for $hl_{query} = [2, 3, 4, 5]$ respectively. $\mathbb{P}[A]\,\mathbb{P}[R \mid A]\,\mathbb{P}[C \mid A \cap R]$ recovers the overall accuracy in Figure 3. Lighter and darker curves denote individual runs and the mean respectively.

Similarly, $B$ is a meaningful event because withholding transitions for a potion pair produces two disconnected halves and applying the withheld transition moves stones between halves, so identifying the correct half is a necessary step to learn the task. By the chain rule, the overall accuracy in Figure 2a is the product of the conditional probabilities:

$$\mathbb{P}[C] \;=\; \mathbb{P}[A]\,\mathbb{P}[B \mid A]\,\mathbb{P}[C \mid A \cap B]$$

Note that according to the definition of $T_{chem}$ and $H_{chem}^{(other)}$, when $\hat{y}_i = y_i$, $C \subset B \subset A$ always holds. We plot these components in Figure 2b. For completeness, we also show $\mathbb{P}[A \cap B^c \mid A] = 1 - \mathbb{P}[B \mid A]$ (red) - predicting the same (incorrect) half $H_{chem}^{(same)}$. We define a *stage* as a training interval characterized by a rise and plateau in the overall model accuracy (Figure 2a) and use the factorized component accuracy to analyze each stage. The stages in Figure 2b are as follows:

1. *Stage 1: In-support (epochs $\approx$20–100).* $\mathbb{P}[A]$ (blue) quickly rises achieving perfect accuracy, indicating that the model quickly restricts its predictions to the valid support set.

2. *Stage 2: Exact match given the correct half (epochs $\approx$100–240).* $\mathbb{P}[C \mid A \cap B]$ (green) quickly rises indicating that the model can identify the target for $q_i$ given the correct half ($\mathbb{P}[B \mid A]$). Yet, the model has not yet learned $\mathbb{P}[B \mid A]$ (at $\sim$ 50% chance accuracy).

3. *Stage 3: Correct half given in-support (epochs $\approx$220–320).* The model completes learning $\mathbb{P}[B \mid A]$ resulting in the complete learning of the task.

We highlight a counterintuitive observation: the exact match accuracy ($\mathbb{P}[C \mid A \cap B]$) rises before the correct half accuracy ($\mathbb{P}[B \mid A]$). This effect is caused by an adjacency bias, where the model exploits the reward structure to identify the set of stones adjacent to the query start stone $x_i$. For a 1-hop transition, $x_i$ has three adjacent stones. For instance, in Figure 1 (left), if $x_i = S8$ is the start query stone with a

+15 reward feature value, the adjacent stones always have a reward feature of +1 (in both the same and other halves). For $x_i = S3$, with reward = -1, the adjacent stones always have a reward feature value of either +1 or -3. This structure allows the model to identify adjacent neighbors without having access to the full latent graph structure.

The transient reduction in the correct half performance ($H_{chem}^{(other)}$ - orange curve), epochs $\approx$ 100-200, confirms this effect. Crucially, two out of three stones adjacent to any $x_i$ reside in the same half (the incorrect half), whereas the true target is located in the other half (the correct half). Therefore, as the model learns to predict the adjacent stones, it becomes biased towards the same half, causing the observed dip in the correct half accuracy. In stage 3, the model correctly learns to predict the correct half, mastering the full task. We provide a detailed empirical analysis of this adjacency effect in Appendix B.

## 4. Composition and Complexity Invariance

We now analyze the composition task, where the model must solve multi-hop queries ($hl_{query} \in \{2, 3, 4, 5\}$) given the support set $S$ of all possible 1-hop transitions $hl_{support} = 1$. As shown in Figure 3, the model exhibits staged learning dynamics across hop lengths while showing no noticeable difference in model performance as hop length increases.

### 4.1. Analytical study of composition learning stages

To study the stages, we define three meaningful events that naturally follow from the composition task structure. Given a chemistry $chem$, query $q_i$, and hop length $k$, we define:

$$\begin{aligned}
A &:= \{\hat{y}_i \in T_{chem}\} & \text{(in-support)},\\
R &:= \{\hat{y}_i \in R_k(x_i)\} & \text{(reachable stones)},\\
C &:= \{\hat{y}_i = y\} & \text{(exact match)}.
\end{aligned}$$

$A$ and $C$ are as defined previously. $R_k(x_i)$ defines the set of stones that can be "reached" by composing k-hop transitions for a given query stone $x_i$. For instance, in Figure 1 (left panel), if $x_i = S1$, and $k = hl_{query} = 2$, the target $y_i$

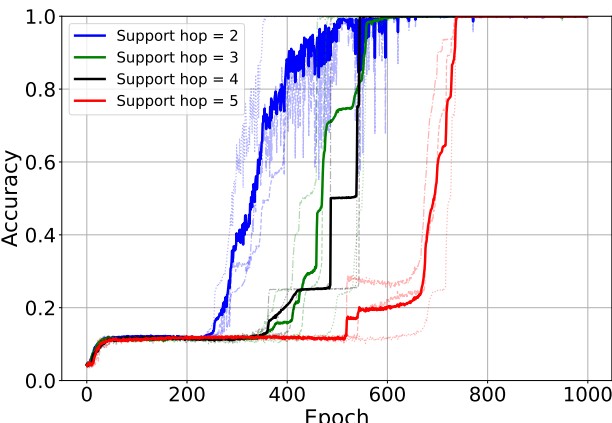

*Figure 5.* Overall decomposition performance. We observe delayed convergence with increasing $hl_{support}$, with 2-hop converging the earliest and 5-hop converging the latest. Lighter and darker curves denote individual runs and the mean respectively.

can only be one of $R_2(x_i) = \{S3, S8, S5\}$. Because, by construction, $y \in R_k(x_i) \subset T_{chem}$, we have $C \subset R \subset A$, and thus,

$$\mathbb{P}[C] = \mathbb{P}[A]\,\mathbb{P}[R \mid A]\,\mathbb{P}[C \mid A \cap R]$$

We now describe the stages in Figure 4.

1. *Stage 1: In-support (epochs $< \approx 30$).* $\mathbb{P}[A]$ (orange) rises rapidly to perfect accuracy, correctly restricting the predictions to the eight support stones. This stage mirrors "in-support" behavior in Section 3.1.

2. *Stage 2: within reachable stones given in-support.* $\mathbb{P}[R|A]$ increases from chance to perfect accuracy as the model identifies the $k$-hop stones reachable from $x_i$. We note that chance levels (purple curves in Figure 4) vary by hop length as the number of reachable stones changes per hop length: $3/8 = 37.5\%$ for $k \in \{2, 4\}$, and $4/8 = 50\%$ for $k \in \{3, 5\}$. Stage 2 is unique to composition, reflecting the need to identify the reachable stones before disambiguating among them.

3. *Stage 3: Exact match given reachable stones.* $\mathbb{P}[C \mid A \cap R]$ reaches near perfect accuracy as the model learns to correctly disambiguate the target among the reachable set for a given $x_i$ and $z_i$ and $k$.

In the composition setting, the query hop length has very little effect on the speed of learning. This suggests that given all 1-hop support transitions, the model can compose them to solve multi-hop queries, at least for $hl_{query} \in \{2, 3, 4, 5\}$.

The sequential nature of the stages suggests a *coarse-to-fine* learning behavior - learning a set of coarse solutions (e.g., in-support and reachable set), before identifying the exact

target. More broadly, our findings are consistent with observations that deep networks learn shared structure before fine-grained distinctions (Arpit et al., 2017; Gopalani & Hu, 2025), and ground these stages in explicit latent structure.

## 5. Delayed Learning in Decomposition

We now identify the difference in model performance when systematically varying $hl_{support} \in \{2, 3, 4, 5\}$ to solve 1-hop queries ($hl_{query} = 1$). Increasing $hl_{support}$ tests the model's ability to infer the underlying chemistry from multi-hop potion sequences to solve a 1-hop query.

We show the decomposition results in Figure 5 and observe that increasing $hl_{support}$ delays model convergence, with $hl_{support} = 2$ reaching convergence sooner than $hl_{support} = 5$. This finding echoes recent works on understanding language model performance when solving tasks with increasing complexity (Dziri et al., 2023; Shojaee et al., 2025), and extends it to a task with a well-defined latent structure. We also observe multiple stages across $hl_{support} \in \{2, 3, 4, 5\}$, which we investigate next.

### 5.1. Analytical study of decomposition learning stages

For a given query $q_i$ containing the stone $x_i$ and potion $z_i$, the target $y_i$ belongs to the set of four stones reachable by $z_i$. We denote this set as the "reachable by $z_i$" because applying $z_i$ can only result in four stones. This is a meaningful event because the end stone state can be extracted by the final potion in $z_i$. For example, in Figure 1 (left), applying a yellow 1-hop potion $x_i = S1$, can only result in a stone from $\{S6, S5, S4, S3\}$ (because yellow can only go to these four stones). We define the sets:

$$A := \{\hat{y}_i \in T_{chem}\} \qquad \text{(in-support)},$$
$$B := \{\hat{y}_i \in H_{chem}^{(reachable)}\} \qquad \text{(reachable by } z_i\text{)},$$
$$C := \{\hat{y}_i = y_i\} \qquad \text{(exact match)}.$$

the multiplication of which gives the exact match accuracy.

$$\mathbb{P}[C] = \mathbb{P}[A]\,\mathbb{P}[B \mid A]\,\mathbb{P}[C \mid A \cap B]$$

We now describe the stages in Figure 6.

1. *Stage 1: In-support (epochs $< \approx 20$).* Across $hl_{support} \in \{2, 3, 4, 5\}$, the model quickly learns to restrict its predictions to the support set, as shown by the rapid rise of $\mathbb{P}_k[A]$ to near perfect accuracy.

2. *Stage 2: Reachable by $z_i$ given in-support.* For $hl_{support} \in \{2, 3, 4, 5\}$, $\mathbb{P}[B \mid A]$ remains at chance ($\approx 50\%$) for many epochs, indicating that the model cannot learn this stage immediately after learning $\mathbb{P}[A]$. This is reflected by the stagnation segment at 12.5% in Figure 5, showing the delayed learning of this stage.

*(a) $hl_{support} = 2$*  *(b) $hl_{support} = 3$*  *(c) $hl_{support} = 4$*  *(d) $hl_{support} = 5$*

*Figure 6.* Staged learning dynamics for decomposition. Components include $\mathbb{P}[A]$ (orange, in-support), $\mathbb{P}[B \mid A]$ (purple, reachable by $z_i$), and $\mathbb{P}[C \mid A \cap B]$ (exact match) shown as blue, green, black, red, for $hl_{query} = [2, 3, 4, 5]$ respectively; $\mathbb{P}[A] \, \mathbb{P} \, [B \mid A] \, \mathbb{P}[C \mid A \cap B]$ product recreates the overall accuracy in Figure 5. Lighter and darker curves denote the individual runs and the mean.

3. *Stage 3: Exact match given reachable by $z_i$.* For $hl_{support} \in \{2, 3, 4, 5\}$, we observe that after learning Stage 2, the exact match accuracy rises quickly, i.e., the model correctly learns the target from the set of four stones. This is reflected in the rapid rise of the stage $\mathbb{P}[C \mid A \cap B]$, from chance to perfect accuracy.

In summary, increasing $hl_{support}$ primarily delays the acquisition of $\mathbb{P}[B \mid A]$, and subsequent model convergence. Nonetheless, the stages are learned in a coarse-to-fine sequence: $\mathbb{P}[A]$ (in-support), followed by $\mathbb{P}[B \mid A]$ (reachable by $z_i$), and finally $\mathbb{P}[C \mid B \cap A]$ (exact match).

To understand why $\mathbb{P}[B \mid A]$ is delayed, we analyzed if other intermediate stages exists, such as initially narrowing the search space in $S$ to a set of candidate stones. However, the simultaneous rise of $\mathbb{P}[B \mid A]$ and other relevant metrics indicated an absence of intermediate stages (Appendix C).

**Sensitivity of Decomposition Learning Dynamics**: While increased task complexity delays latent structure learning, we further observed high sensitivity to hyperparameters in this setting. In Appendix D, we discuss failure modes where certain hyperparameters alter staged learning dynamics or prevent model convergence within the compute budget.

## 6. Plasticity Windows for Staged Learning

A key question arising from staged learning is whether learning later stages depends on continued plasticity in specific layers, i.e. is information for a stage localized in a particular layer or is it distributed? In other words, is plasticity in a layer crucial for learning the subsequent stages, and for how long? We address this question using targeted interventions that remove plasticity (freeze parameters) in individual layers while keeping the rest of the model trainable.

**Intervention and metric:** For a layer $\ell$ and freeze epoch $t_f$, we train normally until $t_f$, after which we freeze the parameters of $\ell$ and keep all other parameters trainable under identical hyperparameters. We apply this intervention to each transformer layer and to the token embedding layer. For a stage $v$, we define stage completion $t_v$ if the

accuracy $a_v(t) \geq \tau$, for a patience window of $P = 3$ epochs. We set the completion threshold $\tau = 0.95$. Thus, $t_v = \min\{t : a_v(t') \geq \tau, \, \forall t' \in [t, t + P]\}$. We measure the intervention effect relative to a control (unfrozen) run as $\Delta t_v(\ell, t_f) = t_v^{\text{freeze}(\ell, t_f)} - t_v^{\text{control}}$, and set $\Delta t_v$ to a maximum value of 1000 if the model fails to converge. We account for negligible variation in delay with a tolerance $\epsilon = 10$. Finally, we express freeze times relative to control stage completion: $\widetilde{t_f} = t_f - t_v^{\text{control}}$ to account for different model initializations. We anchor all results on the completion of the first stage $v = P[A]$ (in-support). Due to compute constraints, we conduct the freezing interventions every 10 epochs only for the withheld potion task (Section 3). Figure 7 shows the mean $\Delta t$ for both $\mathbb{P}[C \mid A \cap B]$ (a) and $\mathbb{P}[B \mid A]$ (b). Results for the individuals runs are in Appendix E.

For both events, freezing transformer layers after $\mathbb{P}[A]$ substantially delays subsequent stage acquisition, and $\Delta T$ decreases with increasing $\widetilde{t_f}$. This demonstrates that continued plasticity is required for several epochs beyond learning $\mathbb{P}[A]$. The stage $\mathbb{P}[B \mid A]$ is more sensitive to early freezing of multiple transformer layers, resulting in large delays or non-convergence within the compute budget (max 1000 epochs). In contrast, freezing early causes shorter but noticeable delays for $\mathbb{P}[C \mid A \cap B]$. Compared to transformer layers, freezing the embedding layer has no noticeable delay ($\Delta t \leq \epsilon$) for $\mathbb{P}[C \mid A \cap B]$ and limited impact on $\mathbb{P}[B \mid A]$. Overall, these interventions reveal that staged learning relies on layer-specific *plasticity windows*, during which a layer must remain trainable for subsequent stage acquisition, and freezing past this window produces no noticeable delays.

## 7. Related Work

**Latent structure learning:** Recent work suggests transformers encode graph-like latent structure during training (Nichani et al., 2024; Henderson et al., 2023), and capture linguistic structure in the BERT model's (Devlin et al., 2019) attention heads (Manning et al., 2020; Ravishankar et al., 2021). However, analyzing the progression of this learning

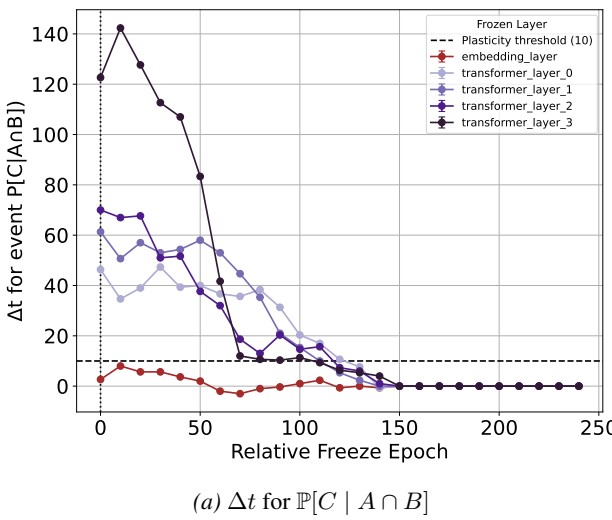

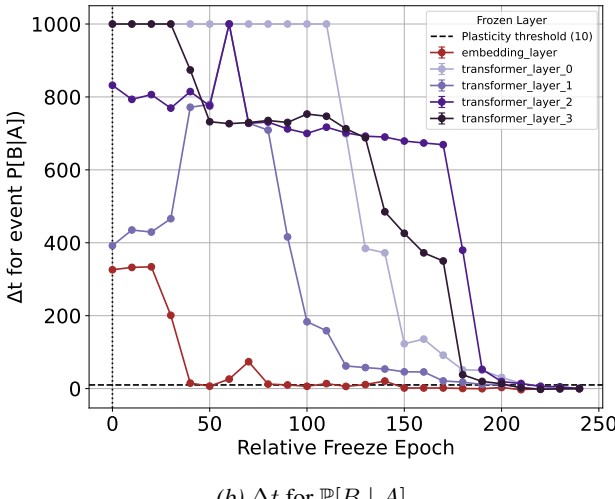

*(a) $\Delta t$ for $\mathbb{P}[C \mid A \cap B]$*

*(b) $\Delta t$ for $\mathbb{P}[B \mid A]$*

*Figure 7.* Plasticity windows under layer-freezing. Mean $\Delta t$ for acquiring (a) $P[C \mid A \cap B]$, and (b) $\mathbb{P}[B \mid A]$ across relative freeze epochs $\widetilde{t}_f$ (anchored to $\mathbb{P}[A]$ completion). The dashed line denotes tolerance $\epsilon = 10$. Large delays from early freezing indicate layer-specific plasticity windows, which are notably much shorter for the embedding layer compared to the transformer layers.

in natural language is difficult due to its complexities. To address this, we use a controlled benchmark to track the learning of specific aspects of latent structure.

Learning in transformers often proceeds in phases or stages (Barak et al., 2022; Siyu et al., 2024; Gopalani et al., 2024; Chen et al., 2024; Gopalani & Hu, 2025; Song et al., 2025; Zhang et al., 2025b) and is frequently linked to 'grokking' (Power et al., 2022; Nanda et al., 2023), and induction heads (Reddy, 2023; Singh et al., 2024; Olsson et al., 2022). Relatedly, staged dynamics have been studied theoretically in simpler gradient-based optimization settings (Saxe et al., 2019; Nakkiran et al., 2019), including recent accounts attributing stages to learning trajectories that evolve near invariant manifolds (Zhang et al., 2025a). While prior works have connected loss plateaus to task features (Barak et al., 2022; Gopalani et al., 2024; Chen et al., 2024), they often lack an explicit latent structure. Dai et al. (2025) investigate graph reasoning via circuit analysis, but do not focus on learning dynamics. We show that the accuracy plateaus correspond to the sequential acquisition of interpretable sub-skills within a known latent structure.

**Composition:** Composition is the ability to combine components in novel ways, and remains a challenge for transformers, particularly with increasing task complexity (Hupkes et al., 2020; Dziri et al., 2023; Petty et al., 2025), and in compositional generalization (Ontañón et al., 2022; Kobayashi et al., 2024). While some works have trained models from scratch on compositional tasks (Thomm et al., 2024), they focus on the final learned model and tasks without a clear latent structure, making it difficult to understand learning behavior. To our knowledge, prior work has not analyzed the latent structure learning dynamics in composition settings.

**Decomposition:** Research on decomposition - the ability to break down complex tasks into simpler components - has recently focused on prompting strategies to improve Large Language Model (LLM) performance (Khot et al., 2023; Pasewark et al., 2024; Wei et al., 2022b; Prasad et al., 2024). However, the staged dynamics of decomposition during latent structure learning remain largely unexplored.

## 8. Conclusion

We investigated the learning dynamics of a small decoder-only transformer on the Alchemy benchmark, using event factorizations to identify discrete acquisition stages. We identified that the model masters different components of the latent structure by first learning a subset of outcomes before refining its predictions to the exact target. In the withheld potion task, the model leverages adjacency biases to prioritize specific targets. We identified a performance asymmetry in tasks with varying complexity: the model is invariant to task complexity in composition, but convergence is delayed with increasing hop length in decomposition, causing prolonged stagnation in specific stages. Finally, the causal interventions reveal that each layer exhibits a plasticity window, during which continued adaptation of the layer is necessary for subsequent stage acquisition.

Future work should examine latent structure learning dynamics in larger models, pre-trained LLMs, or during fine-tuning. Developing more benchmarks and exploring strategies to mitigate decomposition convergence delays remain important directions. Finally, extending this analysis to other architectures such as graph neural networks will help understand the architectural limits of latent structure learning.

## Impact Statement

We contributed towards a better understanding of how a small decoder-only transformer model learns a task with a well-defined latent structure. We showed that the model learns through various stages, each corresponding to different components of the task. We believe our results will advance the field of deep learning by improving the understanding of how the capabilities of transformer models emerge during training. There are many potential societal consequences of our work, none of which we feel must be specifically highlighted here.

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

## A. Training details

We implement all models using PyTorch (Paszke et al., 2019) version 2.6. For hyperparameter tuning, we conduct a large sweep using Weights & Biases (Biewald, 2020) over multiple configurations with the following values for three random seeds. For all experiments, we use the AdamW optimizer (Loshchilov & Hutter, 2019), and 4 NVIDIA L40S GPUs with FP32 precision training. We tune multiple hyperparameters, including the initial learning rate, minimum learning rate, weight decay, and the learning rate scheduler. We show the combinations in Table 1.

| Hyperparameter | Values |
| --- | --- |
| learning rate | [1e-3, 4e-4, 5e-4, 1e-4, 9e-5, 7e-5, 1e-5] |
| scheduler | cosine, cosine w/ restarts |
| weight decay | [0.1, 0.01, 0.001] |
| minimum learning rate (cosine and cosine w/ restarts) | 7e-5, 8e-5, 9e-5, 8.5e-5, 9.5e-5, 1e-5 |

*Table 1.* Set of hyperparameters sweeped over for the experiments.

## B. Reward-Based Analysis for Withheld Potion Pair

To investigate the effect of adjacency bias (Section 3) on the learning curve of $\mathbb{P}[C \mid A \cap B]$, we plot the same events for each individual query stone grouped by reward feature value, $r \in \{+15, +1, -1, -3\}$.

Alchemy has a specific reward structure such that there is exactly one stone state with reward $+15$ and $-3$ per chemistry, and they lie on the diagonal opposites of the cube (see Figure 1, left panel). Furthermore, all three stones adjacent to $+15$ have reward $+1$, and all three stones adjacent to $-3$ have reward $-1$. This defines the reward structure for all 8 vertices of the cube.

From the definition of the events $A$, $B$, and $C$, we can infer that $C \subset B \subset A$, and thus $\mathbb{P}[C \mid A \cap B]$ can be equivalently written as $\mathbb{P}[C \mid B]$. For a query stone, Figure 8a shows the mean accuracy for $\mathbb{P}[C \mid B]$ over all chemistries, grouped by the query stone reward value: $r = +15$ (brown), $r = -3$ (yellow), $r = +1$ (pink), and $r = -1$ (blue). We observe that for $r \in \{-3, +15\}$, $\mathbb{P}[C \mid B]$ rises faster than $\mathbb{P}[C \mid B]$ for $r \in \{-1, +1\}$. Similarly, Figure 8b shows the correct target prediction for the in-support stones, $\mathbb{P}[C \mid A]$, for query start stones $x_r$ with distinct reward values. The learning curves again show faster convergence for $r \in -3, +15$ compared to $r \in -1, +1$. Through the remainder of this section, we explain these observations further by analyzing the reward-based adjacency bias that exists in the underlying Alchemy latent structure.

### B.1. Staged Learning of Withheld Potion Pair Based on Reward-Adjacency

According to Alchemy's underlying latent structure, the reward feature can change by one level per potion transition. For example, from stones with reward $r \in \{+15, -3\}$, transitions are only possible to stones with rewards $+1$ and $-1$, respectively. As a result, stones with rewards $+1$ and $-1$ are adjacent to stones with higher or lower rewards: $\{+15, -1\}$ and $\{+1, -3\}$, respectively.

Consider a query $q = (x, z)$, where $x$ denotes the query start stone, $z$ the query potion, with the target $y$. Let $\mathcal{T}_r$ denote the set of stones reachable from $x_r$ with a 1-hop (single potion) transition. Under the reward constraints above, $\mathcal{T}_{+15}$ and $\mathcal{T}_{-3}$ each contain only three stones with rewards $+1$ and $-1$, respectively. However, for $x_r$ with $r \in \{-1, +1\}$, ambiguity arises: a stone with reward $+1$ may be adjacent to either $+15$ or $-1$, resulting in $|\mathcal{T}_{+1} = 4|$. This ambiguity also exists for a stone with reward $-1$, resulting in $|\mathcal{T}_{-1}| = 4$.

If the model exploits this reward-based structure as a distinguishing signal for predicting the correct half given the query start stone $x_r$, we should see an almost uniform distribution over predictions $\hat{y} \in \mathcal{T}_r$. We test this by plotting $\mathbb{P}_r[\hat{y} = y \mid y \in \mathcal{T}_r]$ for $x_r$ with $r \in \{-3, -1, +1, +15\}$. We show the results in Figure 9 (light blue - dashed), where we observe that during the initial learning period, the values are approximately $\frac{1}{3}$ for $r \in \{-3, +15\}$, and $\frac{1}{4}$ for $r \in \{-1, +1\}$.

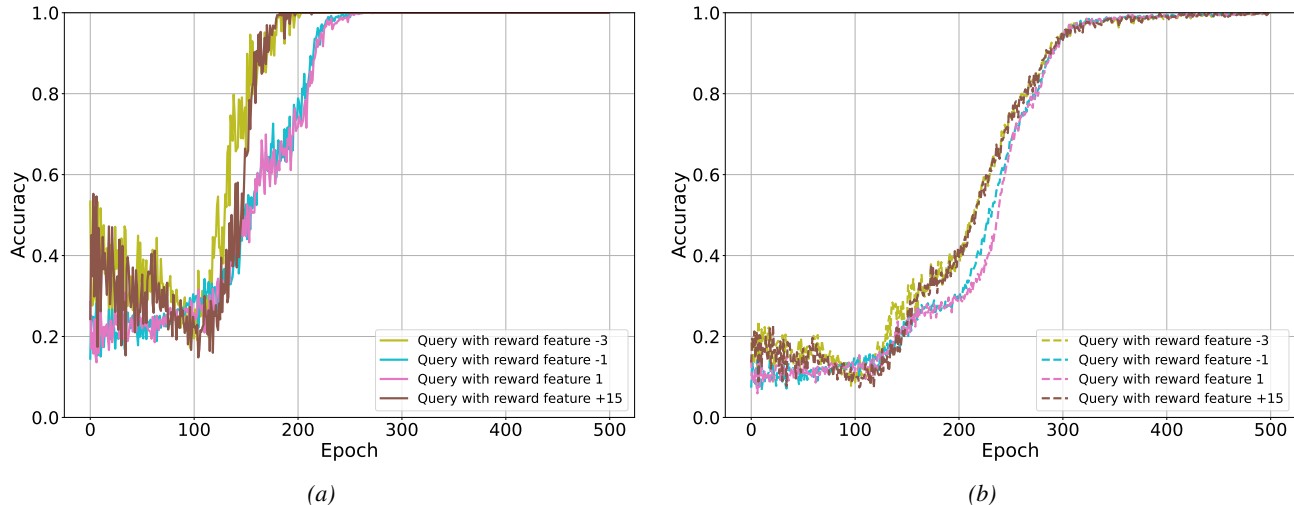

*(a)*          *(b)*

*Figure 8.* Mean prediction accuracy grouped by the reward value of the query. (a) Exact match accuracy given correct half. (b) Exact match accuracy given in-support. We observe that the model learns the exact match accuracy for queries with -3, and +15 reward features faster than those with -1, and +1 reward features.

We also track how the model learns to identify $\mathcal{T}_r$ (within reward adjacent) as the reward-based prediction candidate set within the in-support stones $S$. We measure this by $\mathbb{P}_r[\hat{y}_r \in \mathcal{T}_r \mid \hat{y}_r \in S]$ and show the results in Figure 9 (dark blue). During the initial learning period, for $r \in \{-1, +1\}$, $\frac{|\mathcal{T}_r|}{|S|} = 4/8 = 0.5$, and for $r = \{-3, +15\}$, $\frac{|\mathcal{T}_r|}{|S|} = 3/8 = 0.375$. Note that for stones with $r \in \{-3, +15\}$, the model is initially heavily biased towards predicting $\mathcal{T}_r$, possibly because of the presence of those transitions in the support $S$.

While $\mathcal{T}_r$ captures the reward-based adjacency, the cubic chemistry structure of Alchemy also exhibits geometric adjacency, where each stone has exactly three geometrically neighboring vertices. We denote this by $\mathcal{N}_r$, and plot the corresponding values (within true adjacent, orange) in Figure 9. It naturally follows that for $r \in \{-3, +15\}$, $\mathcal{N}_r$ and $\mathcal{T}_r$ are perfectly superimposed because the geometrically neighboring vertices are exactly the reward adjacent vertices.

We further examine the subset of in-support, non-target reward-adjacent stones denoted as $\mathcal{R}_r \subset \mathcal{T}_r$. $\mathcal{R}_r$ contains the reward-adjacent stones that appear in the support $S$ and are in the same half as that of $x_r$, but are not the correct target. Note that for $r \in \{-3, -1, +1, +15\}$ we have $\mid \mathcal{R}_r \mid = 2$. The red curve in Figure 9 shows this metric as $\mathbb{P}_r[\hat{y}_r \in \mathcal{R}_r \mid \hat{y}_r \in S]$, where the value rises during training despite these stones being incorrect targets.

This mechanism explains the dip in the correct half prediction accuracy observed in Figure 2b (orange) for the withheld potion effect task (Section 3). The model is initially biased towards predicting the target for a while (until $\approx$ epoch 200) according to the reward adjacency structure. This is reflected in the rise of the red curve in Figure 2b. As training progresses, the model unlearns to predict any reward-based adjacent stones randomly, but considers predicting the correct target, which is reflected by the rise in the correct half accuracy.

## C. Investigating the Presence of Intermediate Stages in Decomposition

In the decomposition experiment (Section 5), we observed that increasing $hl_{support}$ delayed the learning of $\mathbb{P}[B \mid A]$ stage. To further investigate this effect, we examined the presence of other intermediate stages.

For the chemistry in Figure 1 (left panel), we define $\mathcal{N}(x_i)$ as the set of immediate 1-hop neighbors of the query start stone $x_i$.

In addition to the events defined in Section C, we then define the following event, for a chemistry $chem$:

$$EN := \{\hat{y}_i \in H_{chem}^{(reachable)} \cup \mathcal{N}(x_i)\} \text{ (extended neighborhood)}$$

For a given chemistry $chem$, query $q_i$, start stone $x_i$, and the potion $z_i$, event $EN$ (cyan) shows whether the model has

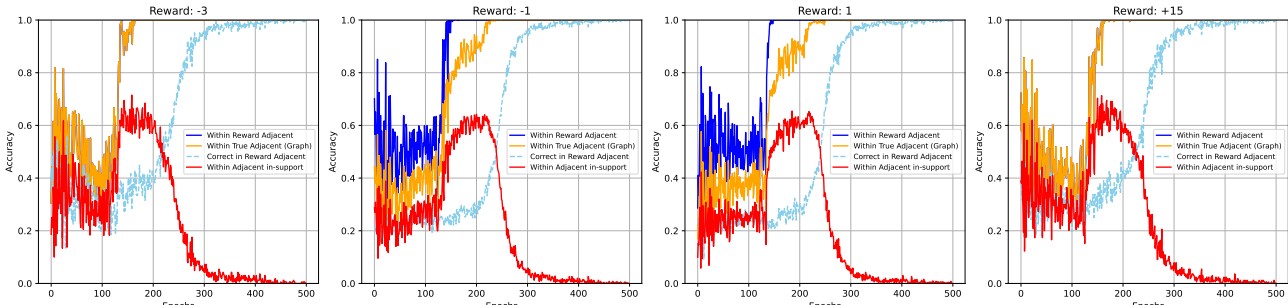

*Figure 9.* Adjacency analysis for each individual reward feature value: i) dark blue shows the learning curve of the adjacent candidate vertices based on only the reward-related latent structure of the 1-hop transitions (correct set of $\mathcal{T}_r$); ii) orange demonstrates the geometrical adjacency ($\mathcal{N}_r$) subset of neighboring vertices, which overlaps the reward adjacency subset, when the start state has reward values of $\{-3, +15\}$; iii) light blue belongs to the learning of the correct target prediction given the reward-based adjacent subset (correct given $\mathcal{T}_r$); iv) red depicts the selection probability of in-support geometrical 1-hop vertices $\mathcal{R}_r$ (only two vertices in the withheld potion task), which are in the wrong half and should not get selected (goes to zero) as the model fully learns the task.

narrowed down its predictions to the search space containing the set of stones reachable from $z_i$, and the stones that are the immediate 1-hop neighbors of $x_i$. Note that for a chemistry $chem$, $|H_{chem}^{(reachable)} \cup \mathcal{N}(x_i)\}| = 6$. The chance accuracy for $\mathbb{P}[EN \mid A]$ is $6/8 = 75\%$. We also plot the ability of the model to narrow down the search space to reachable stones $H_{chem}^{(reachable)}$ conditioned on learning the extended neighborhood for a given $x_i$. We denote this event by $NR$ (neighborhood refinement, pink). The chance accuracy for $\mathbb{P}[NR \mid EN]$ is $4/6 = 66.67\%$. We show the results in Figure 10 for all $hl_{support} = [2, 3, 4, 5]$.

We do not observe any noticeable lag between these events, because the cyan and pink curves rise in immediate succession, indicating a sharp phase transition towards predicting the stones reachable by $z_i$. This indicates that the multi-hop nature of the support $S$ delays the model from successfully extracting the structure of the chemistry, shown by the chance performance of 12.5% in Figure 5 for an extended period of time. This implies that as soon as the model learns that the correct target is in the extended neighborhood, it can almost immediately refine its predictions to the stones reachable by $z_i$.

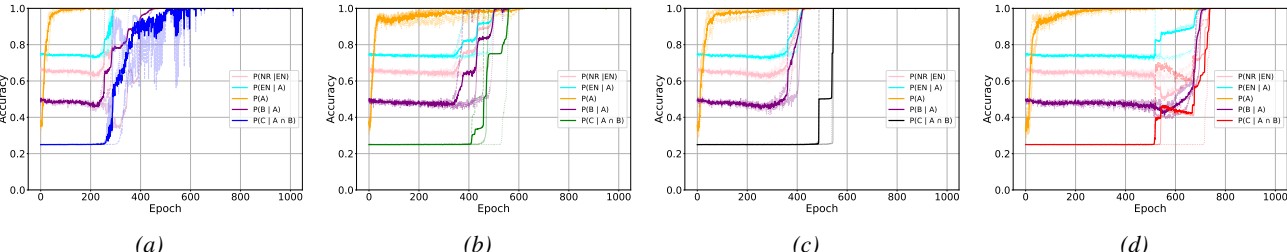

| (a) | (b) | (c) | (d) |

*Figure 10.* Decomposition staged learning dynamics with the extended neighborhood $EN$ curves and neighborhood refinement $NR$ events. We observed that $EN$ and $NR$ rise simultaneously with the reachable by $z_i$ prediction accuracy $\mathbb{P}[B \mid A]$, suggesting the absence of other intermediate stages.

## D. Case Study on Decomposition Hyperparameter Sensitivity

Throughout our analysis, we observed that model performance is highly sensitive to hyperparameter configuration. In this section, we examine how suboptimal hyperparameter choices delay or prevent the observation of certain stages.

**Suboptimal hyperparameters can prevent the observation of stages:** We demonstrate the effect of suboptimal hyperparameters on learning the different components using the 3-hop decomposition task. For a given initialization seed, we show the comparison of the final model performance for the 3-hop decomposition task with different hyperparameter combinations. We show the results in Figure 11. The dark green solid line shows the run from the main results (Section 5), and the other runs show instances with suboptimal hyperparameter values.

We plot the stages for the suboptimal hyperparameter configurations in Figure 12. We highlight that irrespective of the

hyperparameter combinations, the model exhibits staged learning dynamics. Some stages are less affected by suboptimal hyperparameters (e.g., $\mathbb{P}[\mathbb{A}]$ and $\mathbb{P}[B \mid A]$), while other stages are not observed (e.g., $\mathbb{P}[C \mid A \cap B]$). The delay in learning the final stage delays or prevents the final model convergence as observed in Figure 11.

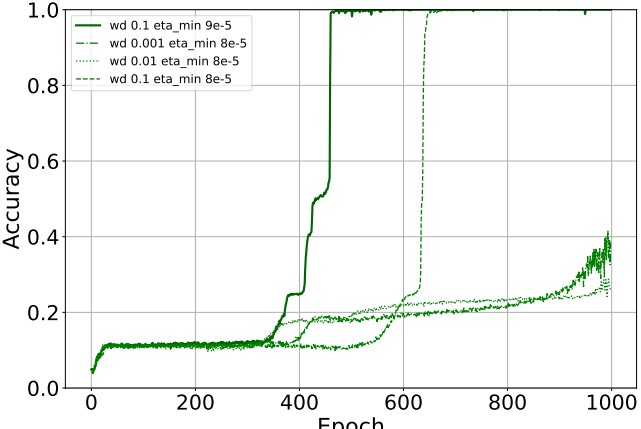

*Figure 11.* Hyperparameter sensitivity results for the 3-hop decomposition task. Suboptimal hyperparameters can delay or even prevent convergence within the compute budget. The solid green curve shows the optimal hyperparameters for a given model initialization. The other runs show model convergence for the same initialization but with suboptimal hyperparameters.

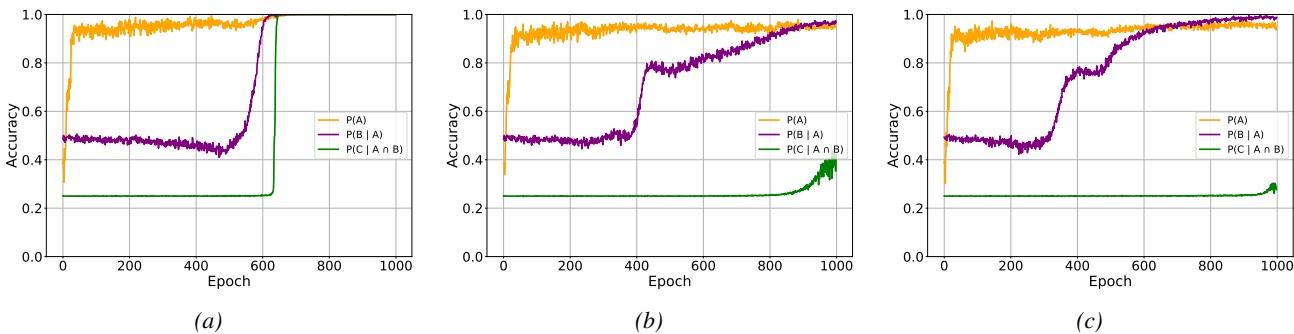

*Figure 12.* Example of hyperparameter sensitivity on the 3-hop decomposition learning stages. (a) Stages with weight decay = 0.1. (b) Stages with weight decay = 0.001. (c) Stages with weight decay = 0.01. In each subfigure, 'orange' denotes in-support prediction $\mathbb{P}[A]$, 'purple' denotes reachable by $z_i$ given in-support $\mathbb{P}[B \mid A]$, and 'green' denotes exact match $\mathbb{P}[C \mid A \cap B]$. Suboptimal hyperparameters can cause the model to delay learning of some stages, resulting in long stagnation segments.

All in all, these results indicate that suboptimal hyperparameters can delay or prevent model convergence by affecting a specific stage.

## E. Additional Results for Plasticity Window

For the withheld potion task, we show the plasticity window results ($\Delta t$ values) for both metrics $\mathbb{P}[C \mid A \cap B]$ and $\mathbb{P}[B \mid A]$, for individual runs in Figure 13.

During the plasticity window, layer-specific freezing interventions cause noticeable delays (or non-convergence) within the compute budget. Transformer layers consistently exhibit longer plasticity windows compared to the embedding layer.

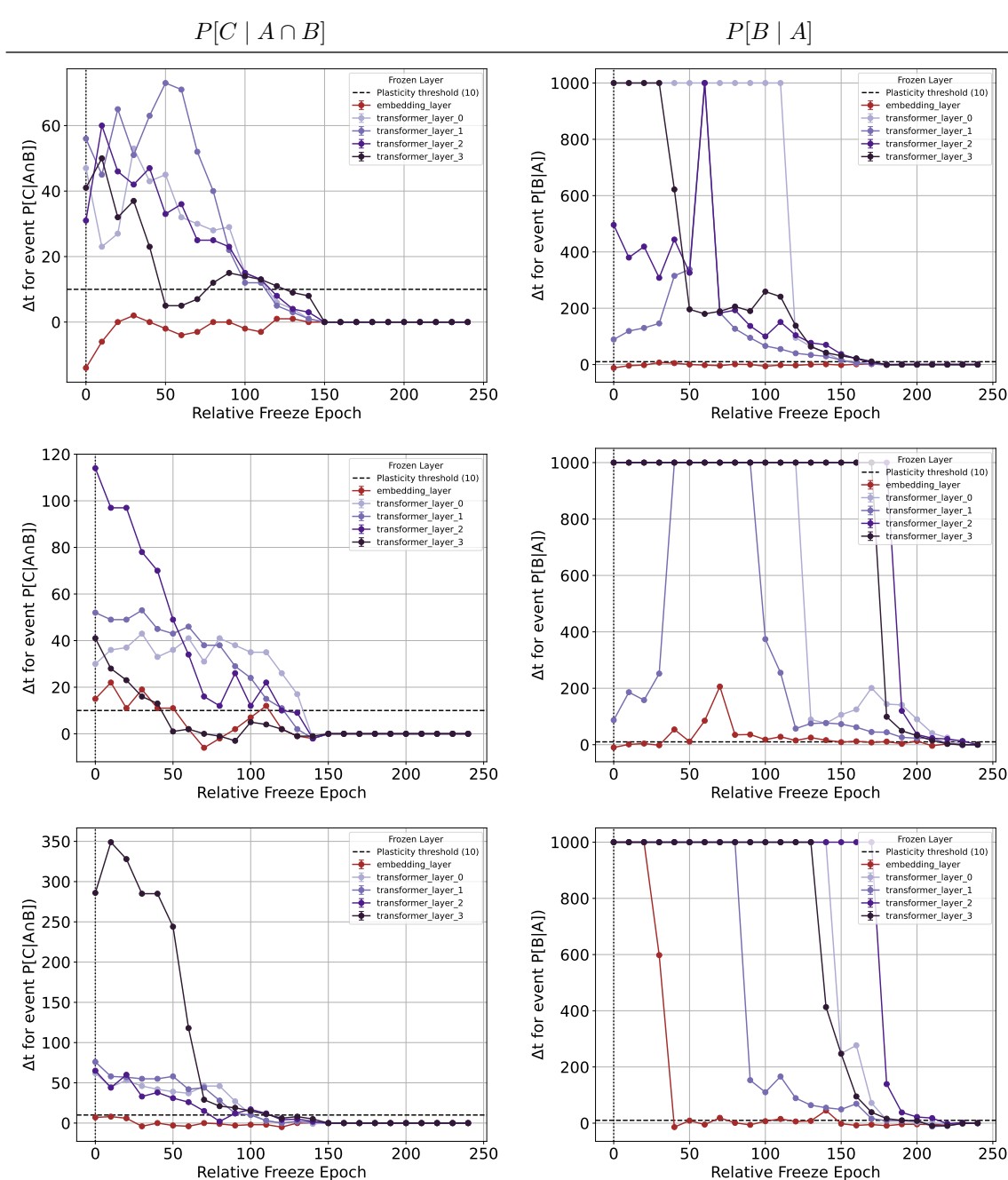

*Figure 13.* Plasticity window results for individual runs on withheld potion effect discovery task. Rows indicate initialization seeds and columns indicate metrics.

