# OpenReview forum: "Understanding the Staged Dynamics of Transformers in Learning Latent Structure"
_ICML.cc/2026/Conference — Submitted to ICML 2026_

### Official Review · Reviewer_t43W · 2026-03-03

**Soundness:** 2
**Presentation:** 3
**Significance:** 2
**Originality:** 3
**Overall Recommendation:** 3
**Confidence:** 4

**Summary:**

This paper studies how small decoder-only Transformers learn latent structure during training, by characterizing **staged learning dynamics** in a controlled setting and tying each stage to interpretable sub-skills and layer-level plasticity. The authors use the Alchemy benchmark: each "chemistry" is an 8-node cubic graph with fixed invariances (paired inverse potions, reward-adjacency constraints). Training is framed as supervised prediction of $p(y \mid q, S)$ from support transitions $S$, with disjoint support and query sets; task difficulty is controlled via hop length of potion sequences.

The paper introduces three tasks that probe different facets of latent-structure learning: (i) **Withheld potion pair**---all 1-hop transitions are given except for one complementary potion pair, and the model must infer the missing effects (the withheld pair splits the graph into two halves that must be aligned). (ii) **Composition**---full 1-hop rules are in the support, and the model answers multi-step (2--5 hop) queries by composing them. (iii) **Decomposition**---only multi-step (2--5 hop) support trajectories are given, and the model must recover 1-hop transitions from longer sequences. To explain plateaus and jumps in accuracy, the authors factor overall accuracy into a product of **interpretable event probabilities** (e.g., in-support $P(A)$, correct subset $P(B\mid A)$, exact match $P(C\mid A\cap B)$), with $C \subset B \subset A$ following from the task structure. These factors define distinct learning stages that progress from coarse constraints to fine-grained prediction. In the withheld task, they observe a counterintuitive ordering (exact match given the correct half rises before correct-half accuracy) and explain it via a **reward-based adjacency bias**: the model first learns to predict adjacent stones, which temporarily biases it toward the wrong half before full structure is acquired.

Composition shows **little sensitivity** to query hop length (2--5), whereas decomposition exhibits **delayed convergence** as support hop length increases, with the bottleneck at learning the "reachable by $z_i$" subset. Finally, the authors run layer-freezing interventions: they freeze a single layer at epoch $t_f$ and measure the delay $\Delta t_v$ in stage completion relative to an unfrozen control, using a formal criterion ($a_v(t) \geq \tau$ for $\tau=0.95$ over a short window). This yields **layer-specific, time-bounded plasticity windows** (transformer layers require longer windows than the embedding layer); interventions are reported for the withheld task only due to compute. The work contributes a controlled framework and an **event-factorization methodology** for studying staged latent-structure learning, plus a first causal-style picture of when and where plasticity is required during training.

**Compliance With Llm Reviewing Policy:**

Affirmed.

**Final Justification:**

Thanks for the detailed follow-up and added experiments. The rebuttal meaningfully addresses several of my questions (run-count denominator, representative failure reruns, inference-time reward ablations, and targeted freezing on decomposition) with *some considerable defects worth examining meticulously*.

**Core remaining issue (soundness):** the “staged dynamics” interpretation is still **not** backed by a fully consistent, run-level evidence chain. In particular, stage-reaching rates for non-convergent runs are estimated via a proxy (thresholding overall accuracy), and the reported rates show **substantial sensitivity** across hyperparameter conditions. This leaves ambiguity about whether the factorized stages reflect a robust learning phenomenon or mainly characterize a subset of successful trajectories under favorable settings.

**Scope limits (significance):** the results are strongest as a controlled diagnostic study (small model, single toy benchmark). In addition, the paper’s “understanding” remains primarily behavioral/task-structured rather than mechanistic localization (e.g., representations/heads/circuits), which limits how far the conclusions can be generalized or interpreted mechanistically.

Overall, the added ablations and plasticity results are valuable, but they do not remove the above constraints. I would reassess with a revision that (i) reports stage completion using the factorized components on a broader set of runs (including failures), and (ii) scopes the claims to match the evidence.

**Key Questions For Authors:**

1. **Failure runs and survivorship bias.** You analyze only runs that reach near-perfect accuracy within the 1000-epoch budget. Could you report success/failure rates per task (and per hop length where applicable) and, for runs that do not converge, characterize where they stall in the factorization—e.g., persistently low $P[B|A]$ vs $P[C|A \cap B]$? Do the same stage decomposition patterns hold qualitatively in failed runs?

2. **Internal mechanism and the “understanding” claim.** Do you have any evidence localizing what changes internally when the model learns $P[B|A]$ (e.g., layer/head attribution, representational probes, attention-pattern shifts, or causal patching)? If not, what is your most concrete hypothesis about which representations implement “correct-half / reachable-set” selection, and how could it be tested?

3. **Reward-adjacency bias: ablations.** The counterintuitive stage order ($P[C|A \cap B]$ rising before $P[B|A]$) is explained by reward-based adjacency. Have you tested whether this order persists if reward is removed from the input features, or if reward assignments are randomized within a chemistry (breaking the reward-adjacency structure)? Such ablations would help corroborate or qualify the proposed mechanism.

4. **Generality of plasticity windows.** Plasticity windows are measured via layer-freezing only for the withheld task (every 10 epochs). Do similar layer-specific windows appear for decomposition (e.g., with a small number of targeted freeze times around Stage-2 onset)? If such experiments are not feasible, would you consider scoping the plasticity conclusion explicitly to the withheld task in the text?

5. **Scope of the composition-invariance claim.** You report that composition learning is largely insensitive to $hl_{\text{query}} \in \{2,3,4,5\}$ given full 1-hop support. Do you have results for $hl_{\text{query}} > 5$, noisier or irregular latent graphs, or larger models? If not, what do you expect to break first (e.g., reachable-set identification vs exact disambiguation), and would you add a brief limitation on the scope of this invariance?

---

From my perspective the paper does not have serious flaws, but I look forward to the authors’ response. If the concerns above—regarding failure-run analysis, mechanistic evidence (or ablations of the adjacency-bias explanation), plasticity-window scope, and the scope of the composition-invariance claim—can be addressed or clearly scoped, **I would consider raising my score**, as these are the key issues currently affecting the persuasiveness and scope of the paper’s conclusions.

**Limitations:**

No. The paper lacks a dedicated Limitations section. While the Conclusion mentions future work (e.g., larger models/benchmarks), it does not explicitly acknowledge key scope limits: (1) analysis restricted to near-perfect runs (potential selection/survivorship effects), (2) plasticity windows measured only on the withheld task, and (3) the small-scale, single-benchmark setting with limited external validation. A short Limitations paragraph stating these points would better align claims with evidence.

**Strengths And Weaknesses:**

**Strengths**

• **Interpretable event factorization with a closed evidence chain.** The decomposition is grounded in task structure: the set inclusion $C \subset B \subset A$ and the chain rule $P[C]=P[A]P[B|A]P[C|A \cap B]$ turn plateaus into testable event probabilities and anchor "stages" (in this controlled setting) to the sequential satisfaction of interpretable constraints (in-support $\to$ correct subset $\to$ exact match). This elevates the analysis from curve-reading to a replicable methodology that could transfer to other controlled settings.

• **Three-task design with a clean composition–decomposition asymmetry.** The setup cleanly separates discovery (withheld potion), forward composition (1-hop support, multi-hop query), and inverse decomposition (multi-hop support, 1-hop query). The resulting asymmetry—composition largely insensitive in convergence behavior to query hop length (2–5), decomposition delayed at $P[B|A]$ with increasing support hop—is well motivated by the setup and directly supports the narrative that composing rules is easier than inferring them from multi-step traces.

• **Structural explanation for the counterintuitive stage order.** In the withheld task, exact match given the correct half ($P[C|A \cap B]$) rises before correct-half accuracy ($P[B|A]$). The authors explain this via reward-based adjacency bias (model first learns adjacent stones; in 1-hop, two of three neighbors lie in the wrong half, temporarily depressing $P[B|A]$). The account is consistent with the Alchemy graph and reward constraints and is backed by finer-grained analysis in the appendix, moving the narrative from phenomenology toward a mechanism-style story.

• **Causal-style intervention with a formal stage-completion criterion.** The layer-freezing experiment uses a well-defined completion criterion ($a_v(t) \geq \tau$, patience, $\Delta t_v$, relative freeze time aligned across seeds), yielding evidence that certain layers must remain plastic during specific time windows for later stages to be acquired—i.e., causal-style evidence rather than correlation alone. The design is clear and reproducible within the withheld task.

**Weaknesses**

• **[Major] Survivorship bias from analyzing only near-perfect runs.** The authors explicitly restrict analysis to runs that reach near-perfect accuracy within the 1000-epoch budget (training protocol / Section 2.3, to focus on dynamics rather than optimization failure). The reported "staged dynamics" may therefore reflect properties of successful trajectories rather than a general learning mechanism. Failure runs—especially in decomposition, where hyperparameter sensitivity and non-convergence are acknowledged (Appendix D)—are excluded, even though they could be most informative about where and why learning stalls (e.g., which conditional $P[B|A]$ vs $P[C|A \cap B]$ fails, and whether the same stage decomposition holds in failed runs). This weakens the inference from "observed stages" to "underlying learning law."

• **[Major] "Understanding" vs. behavioral-level evidence.** The title and goals emphasize understanding latent structure learning, but the evidence remains at the level of behavioral event decomposition ($P(A)$, $P(B|A)$, $P(C|A \cap B)$). The freeze experiment shows that certain layers must remain plastic for later stages but does not identify which representations, heads, or circuits change when the model learns $P[B|A]$ or completes a stage. There is a clear gap between the claim of understanding and the evidence provided; the paper reads more as careful phenomenology plus a plausible structural explanation than as mechanistic understanding. Adding even lightweight representational probes or causal tracing to localize what changes when $P[B|A]$ improves would substantially strengthen the "understanding" claim.

• **[Major] Plasticity-window conclusion may be overgeneralized beyond the evidence.** Layer-specific, time-bounded plasticity windows are established only on the withheld-potion task (freeze points every 10 epochs, per compute constraints); composition and decomposition are not subjected to the same intervention. Stating or implying that "layer-specific plasticity windows" hold for staged learning in general is not supported by the current experiments. Scoping the conclusion to the withheld task or adding at least targeted freeze experiments on the other tasks (e.g., decomposition) would align the claim with the evidence.

• **[Minor] External validity and scope of the "general dynamics" framing.** The setting is deliberately small-scale (~2M params, 4 layers, 8-node cube, full 1-hop support, hop $\leq 5$) to observe dynamics. The finding that composition is insensitive to query hop length may not hold for larger models, noisier or irregular graphs, or longer horizons. This is a matter of claim scope, not of the correctness of the reported phenomena: internal validity is not in question, but the strength of framing the results as general Transformer learning dynamics is limited; reviewers may ask whether the conclusions describe this controlled task or Transformer behavior more broadly.

---

> ### Author Rebuttal · Authors · 2026-03-31
>
> We thank the reviewer for the clear feedback. We address the points below.
>
> **1.** We focused on characterizing successful runs (rather than optimization failures) to attribute stages to the learning of all latent structure components. However, we agree that failed runs are informative. Appendix D shows examples of failed runs where suboptimal hyperparameters for 3-hop decomposition delay / prevent stage convergence rather than eliminating staged learning. Thus, our factorization holds even in failure runs (final accuracy < ~100%).
> We provide the stage success rates for each task below, and will add in the revision.
> |Task|Stage 1 reached|Stage 2 reached|Stage 3 reached|
> |-|-|-|-|
> |Withheld potion pair|100%|100%|~7%|
> |Composition 2-hop|100%|100%|~30%|
> |Composition 3-hop|100%|100%|~48%|
> |Composition 4-hop|100%|~98%|~70%|
> |Composition 5-hop|100%|100%|~30%|
> |Decomposition 2-hop|100%|~98%|~93%|
> |Decomposition 3-hop|100%|~72%|~44%|
> |Decomposition 4-hop|100%|~88%|~30%|
> |Decomposition 5-hop|100%|~37%|~13%|
>
> **2.** Our goal in this work is to analyze the learning dynamics of latent structure, rather than the internal model mechanisms. Our contribution of attributing stages (in-support, correct-half / reachable-set, and exact-match) to latent structures is more fine-grained than prior phase-transition-style analyses [Gopalani and Hu (2025)], but it is still behavioral/task-structured evidence rather than mechanistic. From a mechanistic view, we hypothesize that early layers process coarse grained latent structure, while later layers process fine-grained latent structure. Such hypotheses can be tested using activation patching [Zhang and Nanda (2024)]. We will state this as a natural follow-up in the revision.
>
> **3.** We ran the ablations for the withheld potion task.
>
> A. Randomizing the reward within a chemistry provided inconclusive results as the model did not learn the task (accuracy <~30%). Unfortunately, we cannot tell if this is because the reward signal is no longer useful, or if we introduced a noisy feature (as the original reward mappings are noise free).
>
> B. Removing the reward unfortunately makes the total number of chemistries much smaller, ultimately also reducing the number of unique stones (from 108 to 3x3x3 = 27). For a fair comparison, we ran a baseline with intact rewards but matched the reduced dataset. The model failed to learn the task (~35% accuracy; 5000 epochs).
> So, unfortunately, these approaches have not confirmed the effects of reward on the model performance.
> Despite this, Figure 9 (Appendix B) shows that the model uses the adjacency structure, which explains why $P[C | A \cap B]$ rises before $P[B | A]$. The model learns to predicting stones with rewards that *could* be adjacent (based on the reward value alone) before it figures out to restrict the prediction to the truly adjacent stones.
> Nonetheless, your suggested experiments are a great future direction in settings without data scarcity constraints.
>
> **4.** We acknowledge that the plasticity window results in Section 6 are specific to the withheld potion task, and we will explicitly scope the claims there. We are running the plasticity window experiments for other tasks. We provide the results for composition 2-hop, and 3-hop tasks [here](https://anonymous.4open.science/r/alchemy_latent_structure-C24E/figs/comp_2_3_hop_pw.png). Freezing individual layers does not noticeably delay convergence for ‘reachable-set’, but delays ‘exact-match’. This mirrors Section 6: frozen layers can be compensated by other layers for some stages, but not others. We will add the results for all tasks in the revision.
>
> **5.** We did not study $hl_{query} \gt 5$, but it is possible that performance would eventually degrade with increased hop lengths. We hypothesize that results would follow the pattern we see in the Table provided in point 1 — stages degrade in a reverse-order, with the final stage breaking first, followed by the second, etc. We will add a limitation in the revision and appropriately scope the claim. We did not study noisier or irregular latent graphs as maintaining a fixed latent cubic structure across chemistries was essential for systematic analyses.
> We did not study larger models and the conclusion highlights this limitation. Introducing scale complicates the analyses and it is outside of the scope of this current paper. In the revision, we will motivate the need to study the interplay between model scale and latent structure learning dynamics.
>
> We will add a limitations paragraph clearly scoping the composition complexity claim, motivating scaling experiments, and emphasizing the need for more latent structure datasets.
>
> [Gopalani and Hu (2025)]: Gopalani, Pulkit, and Wei Hu. "What Happens During the Loss Plateau? Understanding Abrupt Learning in Transformers." NeurIPS (2025).
>
> [Zhang and Nanda (2024)]: Zhang, Fred, and Neel Nanda. "Towards best practices of activation patching in language models: Metrics and methods." ICLR (2024)

---

> > ### Author Rebuttal · Reviewer_t43W · 2026-04-01
> >
> > I really appreciate author's work done in rebuttal and additional analyses, which have alleviated my doubts to some extent and strengthened my understanding of the paper. Meanwhile, I still have a few key clarifications/questions that would materially affect my confidence in the **robustness and scope** of the paper’s conclusions:
> >
> > ---
> >
> > 1) **Stage success-rate table: denominator + protocol.**
> >    You report stage-reaching rates (e.g., Withheld Stage 3 reached ≈7%, Decomposition 5-hop Stage 3 ≈13%).
> > **Question**: Please clarify:
> >    (i) how many runs these percentages aggregate over (seeds × HP settings),
> >    (ii) whether they include clearly suboptimal HPs from Appendix D, and
> >    (iii) the exact stage-completion criterion used for this table (same τ/patience as Section 6 or a different one).
> >  *Without this context, it is hard to interpret whether the staged dynamics are broadly robust or mostly a property of a small subset of runs.*
> >
> > 2) **Failure runs: where does learning actually stall in the factorization?**
> >    You state the factorization “holds even in failure runs,” but the current evidence is primarily “stage reached” rates. This does not yet answer *which conditional* fails in non-convergent settings (e.g., persistently low P[B|A] vs. low P[C|A∩B]).
> > **Question**: A minimal addition that would resolve this: show the factorized curves (P[A], P[B|A], P[C|A∩B] or the task-specific analogue) for a few representative failed runs (especially decomposition 3–5 hop), to diagnose the bottleneck.
> > *This directly addresses the survivorship/selection concern.*
> >
> > 3) **Reward-adjacency mechanism: ablations remain inconclusive.**
> >    I appreciate that reward randomization/removal led to non-learning due to data scarcity constraints. Given this, the adjacency-bias account remains plausible but not yet causally corroborated.
> > **Question**: Could you either (a) explicitly scope the claim to “consistent with” (rather than confirmed by ablation), or (b) try a lighter control that preserves dataset size (e.g., masking reward at query time only, or a permutation that preserves reward marginals) to test whether the stage-order effect changes?
> >
> > 4) **Plasticity windows beyond withheld: please prioritize decomposition or scope claims.**
> >    Thanks for adding composition 2/3-hop results. However, the strongest limitation was coverage on the hardest setting.
> > **Question**: If feasible, please provide at least a small targeted freeze study on decomposition (e.g., a few freeze times around Stage-2 onset). If not feasible, explicitly scope the plasticity-window conclusions to the tasks actually intervened upon (withheld/composition).
> >
> > ---
> >
> > Given the above, I acknowledge the progress (especially scoping commitments and added composition freezes), but I would keep my score unchanged just for now and follow author's new evidence. Addressing (1)–(2) would most directly increase my confidence in the generality of the staged-dynamics interpretation.

---

> > > ### Author Response · Authors · 2026-04-08
> > >
> > > We thank the reviewer for the questions and suggestions. We address them below.
> > >
> > > **1.** i. After an initial sweep over all hyperparameters (Appendix A), we fixed the learning rate=1e-4 (for other learning rates, the model failed to reach stage 1 regardless of other hyperparameters). The scheduler had minimal effect; we used cosine scheduler as it performed the best. Then, we ran each experiment for 3 seeds, giving 54 runs per task/hop length combination (3 seeds x 3 weight decays x 6 eta_mins). Tl;dr: denominator is 54.
> > >
> > > ii. Yes, the 54 runs include the suboptimal hyperparameters in Appendix D.
> > >
> > > iii. We initially stored only the final accuracy curves for non-convergent runs. Thus, we estimated stage completion rates by checking if the accuracy crossed the 95% threshold of the following values: 12.5% for $P[A]$; 33% (2,4-hop) or 25% (3,5-hop) for composition $P[B|A]$; 25% for decomposition and 100% for withheld potion $P[B|A]$; 100% for composition / decomposition and 25% for withheld potion $P[C|A \cap B]$. To diagnose bottleneck stages, we reran representative failure runs (point 2).
> > >
> > > **2.** Per request, we reran failure runs for 3,4,5-hop decomposition (see [stages](https://anonymous.4open.science/r/alchemy_latent_structure-C24E/figs/faildec345.png)). We also show withheld potion/composition failure run stages [here](https://anonymous.4open.science/r/alchemy_latent_structure-C24E/figs/ho_cp_fails.png) (final stage stalls consistently across all tasks/hops). For all decomposition hops, in-support stage $P[A]$ is reliably learned the earliest. For 3,4 hops, failure stems from stalling at stage 2 or 3. However, 5-hop decomposition failures show reversed stage ordering — $P[C|A \cap B]$ is learned before $P[B|A]$. We hypothesize that for some hyperparameters, the model possibly becomes trapped in a local minimum by focusing on the reward latent structure.
> > >
> > > Since the reward distribution follows a consistent topological pattern (Figure 1), target stones only have certain reward values (e.g. +15 is always adjacent to +1). Thus, we used two diagnostic tools focusing on the *model's mistakes*: **D1** (Reward plausible error rate: frequency of incorrect predictions with rewards adjacent to query), **D2** (Exact match given prediction is in the reward plausible set). Grouping queries by reward feature (like Appendix B), we show the D1,D2 metrics for 5-hop decomposition [here](https://anonymous.4open.science/r/alchemy_latent_structure-C24E/figs/dec5hopd1d2.png). D1 reaches 100%, but D2 remains close to chance, i.e., the model restricts predictions to plausible rewards but fails to learn the exact target, explaining the reversed stage order. Thus, while the decomposition stage factorization broadly holds with the main findings, 5-hop failures show altered staged behavior driven by reward adjacency (Appendix B). We will add the analysis in the revision.
> > >
> > > **3.** Per request, we conducted **three inference-time ablations on two levels**:
> > >
> > > *Ablations*: **A. Token replacement**: Replacing reward tokens with an unknown \<unk\> token from the vocabulary. **B. Embedding ablation**: Zeroing out the reward embeddings entirely. **C. Attention Masking**: Modifying the attention mask across all layers to block information propagation from reward positions.
> > >
> > > *Levels*:
> > >
> > > *Global*: Applying interventions (A, B, C) to all reward tokens in the input caused accuracy collapse across all methods, confirming reward features are necessary to answer the query.
> > >
> > > *Local*: Applying ablations only to the query reward allowed the model to reach higher accuracy (>40%), but less than 100%. See all ablation results [here](https://anonymous.4open.science/r/alchemy_latent_structure-C24E/figs/wh_abl.png). The in-support stages were still learned the earliest. However, for local ablation, the stage ordering for $P[C|A \cap B]$ and $P[B|A]$ reversed compared to Figure 2b; $P[B|A]$ was consistently higher than $P[C|A \cap B]$ (see [here](https://anonymous.4open.science/r/alchemy_latent_structure-C24E/figs/labl.png)). This corroborates that counterintuitive stage ordering (Section 3) is indeed driven by the reward adjacency bias. We will add the ablations to Appendix B.
> > >
> > > **4.** We now provide plasticity window results for all composition hops [here](https://anonymous.4open.science/r/alchemy_latent_structure-C24E/figs/pwcall.png).
> > > Per request, we conducted targeted plasticity window experiments around stage 2 onset to study stage 3 convergence delay for decomposition *all hops* (see [here](https://anonymous.4open.science/r/alchemy_latent_structure-C24E/figs/pwdecallpba.png)). Consistent with other results, we observed noticeable convergence delay for stage 3 ($P[C|A \cap B]$) when freezing some transformer layers, but no/less delay when freezing the embedding layer. We will add the results in the revision.
> > >
> > > We thank the reviewer for engaging with our work; their suggestions have definitely improved the paper. We hope our responses clarified their doubts.

---

### Official Review · Reviewer_FLJB · 2026-03-11

**Soundness:** 3
**Presentation:** 4
**Significance:** 3
**Originality:** 3
**Overall Recommendation:** 5
**Confidence:** 4

**Summary:**

The paper investigates step-like learning in Transformers on a task designed to have an underlying latent structure (a variation of the Alchemy dataset). The authors consider three variations of the task, which are targeted to evaluate latent structure manipulation, composition and decomposition. They find that introducing task-specific metrics precisely reveal the staged learning dynamics. They additionally find that models can easily compose, but struggle to decompose.

**Compliance With Llm Reviewing Policy:**

Affirmed.

**Key Questions For Authors:**

I am not sure I understand what role the reward is playing in the task. Can the authors clarify this?

**Limitations:**

yes

**Strengths And Weaknesses:**

## Strengths

- The paper is well motivated, and the task that is considered makes a lot of sense to study the staged learning dynamics of latent structure
- The different progress metrics are well justified and offer interesting insights
- Experiments are thorough and most of them are well-thought

## Weaknesses

- This is likely my main concern: the paper reads to some extent as a collection of nice empirical results but is **missing **
   - **a unifying story** (for example the subskills being learned in order should be in that story)
   - **hypotheses for why Transformers are learning things in the way that is reported** (e.g. why are things learned in a specific order, why this order changes between some tasks)
   - **better connections to empirical results in LLMs** (the discussion would benefit from that)\
The paper is currently good; solving these concerns would make it great.
- **It is unclear to me whether the experiments of Section 6 meet the goals that are set in the intro of that section**. If the goal is to understand which layer is implementing which kind of computation, I feel like there might be more direct and conclusive analysis than freezing the learning of specific layers, eg removing one layer / part of that layer, or doing some mechanistic interpretability on the learned model. This section is the one that has most room for improvement.
- there are **a few papers that would deserve to be cited** and to be compared to
   - Allen-Zhu 2025, Physics of LLMs 4.1. The Depo task has some interesting similarities with the composition task considered here, and their findings seem to contradict the ones of this paper: increasing the number of hops makes the task harder to learn (vs. complexity invariance here, cf. Section 4). Musat, ICLR 2025, Wang, Nichani et al., COLT 2025, have results consistent with that. What in the task considered in this paper changes the picture?
   - Zucchet et al. 2025, The emergence of sparse attention: shows that learning sparse attention patterns leads to staged dynamics and provides some understanding on how data affects learning speed. Definitely relevant to that work and might be useful to understand role of data (e.g. "model convergence is delayed as sequence length increases"). It motivates the following experiment: monitoring attention patterns over the course of training and testing whether the different staged observe correspond to the learning of different attention patterns.
- [minor] legends in Figure 4 seem off, B is used in the plots but R in the legend

---

> ### Author Rebuttal · Authors · 2026-03-31
>
> We thank the reviewer for the positive feedback and finding the paper well motivated and having thorough experiments. We provide the responses below.
>
> **1.Unifying Story:** Across all three tasks — withheld potion, composition, and decomposition — our intended unifying story is that a transformer model acquires latent structure in coarse-to-fine manner, focusing on the coarse-grained latent structure acquisition (in-support) first before refining its prediction to the exact target (exact-match).
>
> **Hypothesis for stage ordering:** We hypothesize that the model learns latent structure in a coarse-to-fine manner, because the stages build on each other. It is anomalous for these stages to be learned out of order, which is why we analysed this further in the withheld potion pair task, landing on reward as a reason. While prior work has reported stage transitions (e.g., Gopalani and Hu (2025)), they lack a well-defined latent structure, meaning that stages cannot be attributed to specific latent structure components. We bridge this gap, and will explicitly state this in the revision.
>
> **Connections to empirical results in LLMs:** In the revision, we will expand the related work by connecting our work to and distinguishing it from the broader emergence phenomenon [Wei. et. al. (2022)], and that of circuit formation in large models [Olsson et. al. (2022)].
>
> **2.** We agree that if the goal were to localize the final computation inside the trained model, then more direct post-hoc methods such as layer/component ablations or mechanistic interpretability analyses would be very informative. Our goal for the experiments in Section 6 is narrower. We ask whether continued plasticity in a layer is causally required for subsequent stage acquisition during training, rather than which layer stores the computation for a specific latent structure. We will clarify the ‘stage localization’ wording in Section 6 in the revision. We will also note that attention-pattern analysis, and activation patching are natural follow-up directions for mechanistically characterizing the stages that we attribute to different latent structures.
>
> **3.** Thank you for the papers. We will add comparisons in the revision.
> The DEPO task [Allen-Zhu (2025)] is closely related to our Alchemy composition formulation. In both DEPO and Alchemy, the model is given 1-hop transitions and must answer a multi-hop query. While DEPO formulates this as a directed-permutation task where each node has one successor, Alchemy stones have three outgoing transitions on a cubic graph. Musat (2025) reports increasing hop length degrades performance in a chained retrieval problem. Wang, Nichani et. al. (2025) study a k-fold composition problem with hidden permutations. These settings are different from our composition task.
> We will clarify that different setups may account for different complexity effects, and carefully scope the complexity-invariance claim.
> The phase transition delay finding from Zucchet et. al. (2025) aligns with our decomposition results (increasing $hl_{support}$ delays stage convergence). As our primary focus is to attribute the stages to latent structure components, we will revise the conclusion by stating that monitoring attention patterns and connecting it to latent structure stages is a promising future direction.
>
> [minor]: Thank you for pointing this out. We will update this in the revision.
>
> **Answer to the key question:**
> The original Alchemy dataset was designed for RL settings, where each stone has an associated reward value. An RL agent needs to apply potions to maximize the reward. However, in our experimental setup, ‘reward’ is just another stone feature. We highlight this in Section 2.1 — “Unlike in reinforcement learning settings, where the objective is to maximize reward, here the reward is simply a stone feature.” We have found this confusion arises when explaining our alchemy set up. We will add another reference to the fact that reward is just a feature to other key sections.
>
> [Gopalani and Hu (2025)] Gopalani, Pulkit, and Wei Hu. "What Happens During the Loss Plateau? Understanding Abrupt Learning in Transformers." NeurIPS (2025).
>
> [Wei et. al. (2022)] Wei, Jason, et al. "Emergent abilities of large language models." TMLR (2022)
>
> [Olsson et. al. (2022)] Olsson, et al., "In-context Learning and Induction Heads", Transformer Circuits Thread, 2022.
>
> [Allen-Zhu (2025)]: Allen-Zhu, Zeyuan. "Physics of Language Models: Part 4.1, Architecture design and the magic of Canon layers." NeurIPS (2025).
>
> [Musat (2025)]: Musat, Tiberiu. "Mechanism and emergence of stacked attention heads in multi-layer transformers.” ICLR (2025).
>
> [Wang, Nichani et. al. (2025)] Wang, Zixuang, Nichani, Eshaan et al. "Learning compositional functions with transformers from easy-to-hard data." COLT (2025).
>
> [Zucchet et. al. (2025)] Zucchet, Nicolas, et al. "The emergence of sparse attention: impact of data distribution and benefits of repetition." NeurIPS (2025).

---

> > ### Author Rebuttal · Reviewer_FLJB · 2026-04-01
> >
> > The authors rebuttal will strengthen the paper. In my opinion, this is an interesting paper that will be valuable to the community, and I am therefore keeping my recommendation towards acceptance.

---

> > > ### Author Response · Authors · 2026-04-08
> > >
> > > We thank the reviewer for their time and providing suggestions for improving our work.

---

### Official Review · Reviewer_86yh · 2026-03-12

**Soundness:** 3
**Presentation:** 3
**Significance:** 3
**Originality:** 3
**Overall Recommendation:** 4
**Confidence:** 3

**Summary:**

This paper tries to answer the question "how do transformers learn latent structure?" Towards this goal, the authors analyze a Transformer when trained on the Alchemy benchmark (Wang et al 2021). The authors propose that the controlled setting enables a finer analysis of what the transformer actually learns.

In Alchemy setting, the player starts with a stone, whose properties can be altered by performing various actions ("potions"). A "chemistry" is the set of rules which govern the state-action (stone property, potion) transitions, a chemistry is representable as a directed graph.

Authors train a small decoder-only transformer (2M parameters) to solve novel, multi-step potion sequences. The authors formulate a supervised leaerning problem, where the model is provided with a set of contextual examples ((start state,action sequence,end state) transition triplets) from a specific chemistry, and a query (start state, action sequence) which the model needs to predict the end state. Task complexity is defined via the length of the action sequence. The authors study three variants
1. Inferring a withheld potion pair
2. Compose 1-hop support transitions to solve multi-hop queries:
3. Decompose complex multi-hop support to solve 1-hop queries

In each case, the authors observe that the accuracy of the model increases in stages. In addition, authors try to analyze what each stage represents in terms of the underlying latent structure learned. In addition:
- in task 2, the query hop length has little effect on speed of learning, suggesting that the model can compose 1-hop transitions effectively, regardless of query hop length.
- in task 3, the authors observe that learning becomes slower as support hop length increases.

Finally, the authors also study the effect of freezing transformer layers after certain stages, to understand if information for a stage are localized in particular layers.

**Compliance With Llm Reviewing Policy:**

Affirmed.

**Final Justification:**

The rebuttal addressed my technical concern about the analysis. I still have some reservations about the generality of conclusions to problems beyond alchemy, but I do feel that the authors did a careful job analyzing the specific alchemy problem, which may be an useful step towards understanding more general problems.

**Key Questions For Authors:**

see weaknesses

**Strengths And Weaknesses:**

## Strengths
Understanding how Transformers learn latent structure is an important question.

The problem is well-formulated, and I am somewhat convinced by the authors' analysis. The decomposition of transitions by conditional probability appears to align well with the staged accuracy jumps with training epoch. This gives an interesting insight into how transformers are able to learn latent structure.

## Weaknesses
- The Alchemy-specific notation (stone, potion) makes the paper unnecessarily hard to follow. It seems that the problem is exactly frame-able as a tabular MDP problem, and the task-formulation reduces to in-context learning of the transition function in tabular MDP. This point is mainly a criticism of the presentation, not the content, of the paper.
- The scope of analysis is limited to the specific Alchemy setup. This really calls into question the generalizability of their conclusions. This is particularly problematic because "Alchemy" is not a commonly-studied problem. It would be much better if the authors can generalize their findings across 1-2 other similar problems. Alternatively, the authors should at least try to justify how the Alchemy is a generally interesting task.
- Section 6 on plasticity window is very light on details. What transformer layers are being frozen per stage? How did the authors decide what layers to freeze for each stage?
- Relatively minor (does not influence my score): The authors should do a more comprehensive comparison to related work. Particularly relevant are recent papers that try to mechanistically understand in-context learning, in-context RL, and chain-of-thought.

---

> ### Author Rebuttal · Authors · 2026-03-31
>
> We thank the reviewer for providing a good summary, highlighting that the problem is important to study, and noting that our analyses are convincing. We believe addressing your points will make the paper better and improve the presentation of our analyses. We respond to each question below.
>
> **1.** Thank you for this helpful presentation suggestion. We agree that the Alchemy chemistries can be also be framed as a finite/tabular MDP: each chemistry defines a small transition system over a finite set of states (stones), and the support set provides transition examples ($x_s, z_s, y_s$), from which the model must infer the chemistry-specific transition rule to answer a query. We used the original Alchemy terminology (“stone”, “potion”, “chemistry”) primarily to stay faithful to the benchmark and related work; our event-based factorization naturally aligns with the benchmark’s language. Our framing aligns with relevant work studying in-context learning in transformer models (e.g., Singh et. al. (2024); Mittal et. al. (2025)). In the revision, we will add the sentence in Section 2.2 to describe an alternative tabular MDP framing. We hope this will improve the clarity of the paper for some audiences.
>
>
> **2.** It is true that the current analysis is specific to Alchemy, and we will clarify the scope of our claims more explicitly. We chose depth over breadth intentionally — controlled settings are often how deep understanding is first established. However, our findings align with patterns seen in broader settings.  For example, recent works [Singh et. al, (2024); Gopalani et. al., (2024); Gopalani and Hu (2025)] demonstrate phase changes across multiple tasks. Our paper adds to the community’s understanding by diving deeper into staged dynamics to show exactly which latent structure is associated with the underlying stages. This fine-grained attribution is only possible by restricting the setting, but it lays the groundwork for future work expanding into other similar domains.
> In the revision, we will add a sentence in the Introduction to motivate the accessible nature of Alchemy’s latent structure and why it provides a controlled test bed for studying latent structure learning dynamics.
>
>
> **3.** We thank the reviewer for this point. To clarify, we studied the effect of freezing all the layers. We froze each transformer layer separately (freezing all components inside a transformer block) and the embedding layer to study how restricting learning affects subsequent stage acquisition. The subfigures in Figure 7 show the plasticity windows for each layer (embedding, and each of the four transformer layers).  We froze each layer after the first stage (P[A] in-support) crossed $\tau = 0.95$.
> In the revision, we will add more details in Section 6 and clearly list the frozen layers.
>
> **4.** Thank you for suggesting a comparison to other relevant works. Based on other reviewer feedback we will add comparisons to the mechanistic interpretability literature (e.g., Zhang and Nanda (2024); Yang et. al. (2025)), and motivate studying the effect of chain-of-thought [Wei et. al. (2022)] on the latent structure acquisition in the decomposition task. We would be glad to add comparisons to any other suggested papers to further strengthen our work.
>
>
>
> [Singh et. al. (2024)]: Singh, Aaditya K., et al. "What needs to go right for an induction head? a mechanistic study of in-context learning circuits and their formation." ICML (2024)
>
> [Mittal et. al (2025)]: Mittal, Sarthak, et al. "Does learning the right latent variables necessarily improve in-context learning?." ICML (2025)
>
> [Gopalani et. al. (2024)]: Gopalani, Pulkit, Ekdeep S. Lubana, and Wei Hu. "Abrupt learning in transformers: A case study on matrix completion." NeurIPS (2024).
>
> [Gopalani and Hu (2025)]: Gopalani, Pulkit, and Wei Hu. "What Happens During the Loss Plateau? Understanding Abrupt Learning in Transformers." NeurIPS (2025).
>
> [Zhang and Nanda (2024)]: Zhang, Fred, and Neel Nanda. "Towards best practices of activation patching in language models: Metrics and methods." ICLR (2024)
>
> [Yang et. al. (2025)]: Yang, Yukang, et al. "Emergent symbolic mechanisms support abstract reasoning in large language models." ICML (2025)
>
> [Wei et. al. 2022]: Wei, Jason, et al. "Chain-of-thought prompting elicits reasoning in large language models." NeurIPS (2022)

---

> > ### Author Rebuttal · Reviewer_86yh · 2026-04-03
> >
> > I will increase my score to weak accept.

---

> > > ### Author Response · Authors · 2026-04-08
> > >
> > > We thank the reviewer for their time and providing valuable feedback on our paper.

---

### Official Review · Reviewer_JPnE · 2026-03-13

**Soundness:** 3
**Presentation:** 4
**Significance:** 2
**Originality:** 4
**Overall Recommendation:** 4
**Confidence:** 3

**Summary:**

The paper investigates the learning dynamics of transformers on the Alchemy benchmark. The authors introduce three different tasks based on the Alchemy dataset and show that the model masters different components of the latent structure. In addition, they identify layer-specific plasticity windows during which freezing substantially delays or prevents stage completion, offering a detailed view of how capabilities
evolve during training.

**Compliance With Llm Reviewing Policy:**

Affirmed.

**Final Justification:**

My concerns have been addressed. I will raise my score to 4.

**Key Questions For Authors:**

See Weaknesses.

**Limitations:**

The authors have discussed the limitations.

**Strengths And Weaknesses:**

Strengths:
1. The paper is well-written and easy to understand. Using the illustration of the chain rule, it clearly explains the different components.
2. The topic is novel and interesting. It analyzes staged dynamics of transformers based on a special Alchemy dataset.
3. The paper excels in presenting experimental results with clarity and richness.

Weaknesses:
1. The paper is restricted to a specific task and a fixed structure of the model with eight stones and four hidden layers. In order to generalize to other fields, it's better to extend to more complicated settings.
2. The reachable stones in Section 4 and Section 5 seem different. In the composition task, the reachable set is mainly based on the starting stone. In the decomposition task, the reachable set is obtained from the potion, which is unrelated to the starting point. The authors could disscuss more about the difference.
3. The robustness of the composition task may be dependent on the data structure. Composition in Alchemy is based on a fixed graph, which is close to a deterministic computational problem. In contrast, real-world composition is far more complex. The paper could dicuss about this.
4. As explained in Section 3, the observation that the exact match accuracy rises before the correct half accuracy is caused by an adjacency bias. It's an interesting observation, but the transition rule relies entirely on the designed reward feature that is highly domain-specific. It fails to provide generalizable insights for understanding transformer learning dynamics on other types of data.
5. In Figure 5, the 2-hop curve is oscillating, while the other curves are relatively smooth. What explains this behavior?

---

> ### Author Rebuttal · Authors · 2026-03-31
>
> We thank the reviewer for acknowledging our paper’s clarity, novelty, and the rich presentation of the experimental results. We respond point-by-point below and will incorporate these clarifications in the revision.
>
> **1.** It is true that our study is in a controlled setting: we use only the complete chemistries from the Alchemy benchmark, and a small decoder-only transformer — this is intentional. We chose Alchemy because its latent structure is well-defined and manipulable, which allows us to design systematic experiments where stages can be precisely attributed to latent structure. **Attributing the stages to different latent structure components is much harder to do in natural language or less controlled domains. Accordingly, our intended claim is not that the exact dynamics shown here immediately generalize to all settings, but that Alchemy provides a clean testbed for studying latent-structure acquisition in transformer models in a way that is difficult elsewhere.**
> Coming to model scale: there is likely interplay between model size and structure learning, but our goal was not to study that interplay. We chose to keep our paper focused and instead pursued a deeper analysis of a specific model scale (~2M). We will further emphasize the need for scaling experiments in the revision.
>
> **2.** The reviewer is correct: there is a difference in how the reachable stones in composition and decomposition are defined. In the composition setting (Section 4), the query is a multi-hop sequence starting from a specific stone $x_q$, so the natural intermediate stage is whether the prediction lies in the set of stones reachable from that start state after $k$ hops, i.e., $R_k(x_q)$. In the decomposition setting (Section 5), the query is 1-hop, while the support consists of multi-hop transitions.  Multi-hop transitions obfuscate the intermediate stones, but they do surface a final potion, and the final stone in the sequence. This allows the model to infer which potions lead to which stones. In other words, the relevant learning stage is defined as the stones reachable by the query potion $z_q$. This is also equivalent to having a set of four stones belonging to a half of the chemistry, similar to the withheld potion pair experiment. We will clarify the reachable stones definition between composition and decomposition in Sections 4 and 5.
>
> **3.** In the Alchemy setting, the ‘transitions’ are deterministic under a fixed graph setting, which may not be true in real-world situations. We acknowledge that this is a simplification, but we note that using the controlled Alchemy setting provides us with a clean way of studying latent structure learning dynamics in a composition setting, and attributing stages to different latent structures. Such an analysis is difficult in less controlled settings, where it is also harder to systematically vary complexity (e.g., hop length).
> In Section 1, we motivated Alchemy as a controlled setting in contrast to natural language. In the revision, we will expand this motivation by contrasting it to real-world settings and motivate the need for studying latent structure learning dynamics in more complex settings.
>
> **4.** It is true that the adjacency bias in Section 3 arises from a reward structure property specific to Alchemy. While the specific stages are task-dependent, our results are revealing precisely because our controlled setting allows us to attribute it directly to a specific latent structure component, which is something that is not possible in less controlled settings where the latent structure is not well-defined. Prior work has reported staged dynamics in transformers (e.g., Gopalani and Hu (2025)), but has not attributed those stages to the acquisition of specific latent structure components, as the tasks do not have a clear latent structure. We address this gap by showing that, across multiple tasks, the learning stages correspond directly to the model acquiring distinct latent structure components, providing a fine-grained view of the learning dynamics that is currently absent in prior work.
> In the revision, we will highlight this contribution explicitly and connect it better to prior work.
>
> **5.** This oscillating behavior for the 2-hop decomposition task in Figure 5 is due to hyperparameter sensitivity. Given the set of possible hyperparameter configurations (Appendix A), we selected the runs that converged the earliest (best runs across multiple seeds for each hop considered independently). We show an example for the 2-hop decomposition task (that converge later) not showing this oscillating behavior in this [plot](https://anonymous.4open.science/r/alchemy_latent_structure-C24E/figures_alchemy_misc/March_30_decomposition_2_hop_individual_runs_aXXXX-7uracy.jpg).
>
> [Gopalani and Hu (2025)]: Gopalani, Pulkit, and Wei Hu. "What Happens During the Loss Plateau? Understanding Abrupt Learning in Transformers." Advances in Neural Information Processing Systems (2025).

---

> > ### Author Rebuttal · Reviewer_JPnE · 2026-04-04
> >
> > Thank you for your response. I will raise my score to 4.

---

> > > ### Author Response · Authors · 2026-04-08
> > >
> > > We thank the reviewer for taking the time to review our work and providing valuable feedback.

---

### Decision · Program_Chairs · 2026-04-30

**Decision:**

Reject

**Comment:**

This paper focuses on a synthetic setting in which it is possible to analyze the granularity of individual subtasks within an objective. This paper is pretty restricted to the synthetic setting, where they argue that the model shows evidence of coarse to granular learning. It’s clear that’s what’s happening in the synthetic setting but less clear how general the phenomenon is. However, there is existing evidence of similar phenomena in other settings, which makes me sympathetic to the idea.

Given existing examples of these types of staircase learning phenomena both in toy settings (eg, in Barak et al., which is cited, and in other work from Eran Malach and in the work of Emmanuel Abbe, which is not) and in practice in MLMs (eg, in Chen et al., which is also cited), the main contribution here is probably the finding that layer specific freezing prevents the model from completing a particular stage, therefore localizing where each stage of learning occurs.

The idea is promising, but the synthetic setting is very restrictive and weak. One thing that would improve the paper is to leverage one of the other known settings, outside of the alchemy setup, to show that the causal findings still apply. I find that the paper does contribute to the accruing evidence for multistage learning dynamics by linking them to localized stages, but I encourage the authors to emphasize the novel contributions in future revisions.